# **Benchmarking Catchment-Scale Snow Water Equivalent Datasets** and Models in the Western United States

Ethan Ritchie<sup>1</sup>, Andrew W. Wood<sup>1,2</sup>, Ryan Johnson<sup>3</sup>, Adrienne Marshall<sup>1</sup>, Josh Sturtevant<sup>1</sup>, Dane Liljestrand<sup>3</sup>, Emily Golitzin<sup>3</sup>

<sup>1</sup>Colorado School of Mines, Hydrological Science and Engineering, Golden, CO USA
<sup>2</sup>NSF National Center for Atmospheric Research, Terrestrial Sciences Section, Boulder, CO USA
<sup>3</sup>University of Utah, Civil and Environmental Engineering, Salt Lake City, UT USA

Correspondence to: Andrew Wood (awwood@mines.edu), Ethan Ritchie (ritchie.ethan.m@email.com),

Abstract. This study benchmarks a wide range of snow water equivalent (SWE) models and data products at a catchment scale in the Western US, and discusses an experimental protocol to facilitate community-wide intercomparisons. Utilizing lidar-based ASO (Airborne Snow Observatory) SWE estimates as a 'ground truth', this study evaluates the performance of multiple SWE products, including SNODAS, SWANN (4km and 800m), the US National Water Model (NWM), UCLA-SWE, SWEMLv2, NLDAS-2 (VIC, Noah, and Mosaic), ERA5-Land, Daymet and the CONUS404 dataset. We use SWE aggregated to hydrologic catchments as the standard spatial basis for assessment, focusing on multiple spatially-variable performance metrics. UCLA-SWE, SWANN (both 800 m and 4 km), and SWEMLv2 show the strongest agreement with ASO SWE, each achieving Kling-Gupta Efficiency (KGE) values above 0.6. SNODAS also performs competitively with these higher-performing models. The coarser-resolution products generally perform poorly at the catchment scale. Notably, ERA5-Land and the NLDAS-2 Mosaic and VIC models demonstrate strong skill for basin-average SWE (R<sup>2</sup> > 0.9), while the NLDAS-2 Noah model exhibits weak performance across both spatial scales. Noting the lack of a common community standard for SWE product and model evaluation, we use the results of the multi-dataset analysis to explore potential experimental protocols for a standardized SWE evaluation that could support community-wide intercomparison and benchmarking of existing and new SWE products. SWE datasets are a critical component in hydrologic prediction practices such as water supply forecasting, thus the use of experimental standards proposed herein could facilitate quantitative guidance for agency and stakeholder adoption of specific SWE products in decision support applications.

#### 25 1 Introduction

Snow is a critical component of freshwater resources, storing water seasonally and releasing it gradually to sustain streamflow, groundwater recharge, and water supply for ecosystems, agriculture, and communities. Snowmelt volumes can often be predicted with usable accuracy, making snowpack estimates central to water resource management, but its quantity and distribution are highly sensitive to climate variability, complicating accurate measurement and forecasting (Natural Resources Conservation Service, 2010). In the western United States (US), snow water equivalent (SWE) has traditionally

55

been measured through ground-based networks, most notably the Snow Telemetry (SNOTEL) system (Serreze et al., 1999), which provides automated in situ measurements of mountain snowpack. While valuable, point-based observations do not measure spatial variability across complex terrain, motivating the development of areal SWE products derived from remote sensing, reanalysis, and modeling approaches.

SWE estimation methods can be categorized into four main approaches, which differ in temporal and spatial resolution, underlying assumptions, and data sources: physically-based models, reanalysis products, data-driven models, and SWE derived directly from satellite remote sensing imagery. Physically-based models, such as those used in SWE products from the Daymet dataset (Thornton et al., 2021), NLDAS-2 (Xia et al., 2012) land surface models (LSMs), and the US National Oceanic and Atmospheric Administration (NOAA) National Water Model (NWM; Cosgrove et al., 2024), simulate snow 40 accumulation and melt using representations of land surface energy and water balances, ranging from simple temperatureindex formulations to full energy-balance models with data assimilation. Reanalysis products, including Western United States UCLA Daily Snow Reanalysis (UCLA-SWE; Margulis et al., 2019), the Snow Data Assimilation System (SNODAS; Carroll et al., 2001), and the fifth generation European ReAnalysis land component (ERA5-Land; Munoz-Sabater et al., 2021), combine observations and model outputs to generate spatially complete, physically consistent hydroclimate variable records. Notably, UCLA-SWE assimilates remotely sensed fractional snow-covered area (fSCA) derived from Landsat and the Moderate Resolution Imaging Spectroradiometer (MODIS; Riggs et al., 2017), enabling improved characterization of seasonal snow in regions with sparse in situ data, such as high-elevation catchments. Data-driven models, such as Snow Water Artificial Neural Network (SWANN; Broxton et al., 2016) and Snow Water Equivalent Machine Learning (SWEML; Liljestrand et al., 2024), leverage historical observations and machine learning, training on observational datasets such as SNOTEL and the Parameter-elevation Regressions on Independent Slopes Model (PRISM; Daly et al., 1994), with 50 performance dependent on input data quality and coverage. Remote sensing-based methods, including high resolution Airborne Snow Observatory (ASO) lidar SWE (Painter et al., 2016) and satellite passive microwave products (e.g., SMMR, SSM/I, AMSR), provide indirectly derived SWE estimates at high or coarse resolution, respectively.

Aerial (airplane-based) lidar (Light Detection and Ranging) measurements of snow depth have been collected since 2013 by the Airborne Snow Observatory (ASO), a collaboration initially led by NASA's Jet Propulsion Laboratory, California Department of Water Resources, and other state and local entities, that has since privatized (ASO Inc.). ASO provides high-resolution (50 m) SWE estimates derived from lidar-based snow depth and modeled snow density (Painter et al., 2016). It's important to recognize that ASO SWE is not a direct observation (due to its combination of lidar-based depth and model density), and it presents temporally discontinuous snapshots of SWE (with typically 1-5 scenes obtained in a basin per year). Notwithstanding these limitations, it is among the most detailed spatially distributed quasi-observational SWE datasets available. Researchers have duly used ASO datasets to assess remotely sensed snow (Behrangi et al., 2018; Broxton et al., 2024), benchmark new SWE estimation methods and models (Dawson et al., 2018; Margulis et al., 2019; Oaida et al., 2019), train data-driven models (Liljestrand et al., 2024), evaluate basin precipitation (Cao et al., 2018; Henn et al., 2016), and support other SWE intercomparisons (Yang et al., 2023). In particular, Yang et al. (2023) compared five SWE products over

the western US, including two reconstruction products (REC-ParBal; Bair et al., 2016, REC-INT; Guan et al., 2013), a reanalysis-based product (REC-DA/UCLA-SWE), and two operational products (SNODAS and the National Water Model). They found that UCLA-SWE (REC-DA) achieved the highest overall accuracy and best captured the spatial distribution of SWE, while SNODAS and NWM exhibited substantially lower accuracy. Retrospective approaches (REC-INT, REC-ParBal, and REC-DA) generally outperformed real-time SWE estimates, likely due to the additional information available to the retrospective models.

Recent global intercomparison efforts highlight the persistent challenges of accurately representing SWE, especially in complex terrain. Mudryk et al. (2025) evaluated 23 gridded SWE products using an expanded reference dataset combining snow course and airborne gamma observations. Their analysis found that ERA5-Land and the Crocus snow model (Brun et al., 2013) versions performed best overall, benefiting from finer spatial and vertical resolution and the absence of surface snow assimilation. Most products captured SWE climatology and variability in low-relief regions but performed poorly in mountainous areas, emphasizing the ongoing need for targeted high-resolution SWE products. Complementing these global and regional efforts, this study focuses on the mountainous western US, emphasizing scales relevant to basin- and catchment-level water management. We evaluate both global and regional SWE datasets against catchment-level ASO SWE (a high-resolution observational reference), using scenes from over 400 ASO flights across multiple basins.

The creation of a standardized protocol for evaluating SWE products (e.g., with a common spatial/temporal basis, reference observation, and evaluation metric convention) can be valuable for both research and operational communities, providing evidence-driven insights into the performance of new and existing methods. The potential utility of such standardized intercomparison has been strongly demonstrated previously for streamflow, notably with the widespread adoption of the Catchment Attributes and Meteorology for Large-Sample Studies (CAMELS; Addor et al., 2017; Newman et al., 2015) datasets. CAMELS defined a collection of 671 watersheds in the US and provided the associated hydrometeorological timeseries, geophysical attributes, and benchmark model simulation results. Since its introduction as a benchmarking reference for streamflow simulation (Newman et al., 2017), CAMELS has become the basis for assessment in many streamflow modeling papers (e.g., Frame et al., 2022; Klotz et al., 2022; Kratzert et al., 2019; Rakovec et al., 2019; Xie et al., 2021), the focus of entire 'large-sample hydrology' conference sessions, and has also spawned efforts to extend the original contiguous US (CONUS) CAMELS datasets globally (Kratzert et al., 2023). Such datasets enable direct comparisons between diverse modeling results (from process based and machine learning techniques) that have previously been difficult to reconstruct from individual studies due to a lack of standard experimental conventions (such as testing periods, metrics and locations).

This study aims to expand on the success of the CAMELS community benchmarking activities in developing a similar focal point for SWE estimation, given that there are now over a dozen spatial SWE products that cover (at least) the extent of the CONUS. In particular, we (1) evaluate the performance of multiple SWE estimation methods with respect to ASO SWE observational estimates (referred to as 'observations' or brevity); and (2) discuss and propose potential common experimental protocols for evaluating SWE that can be used to assess existing and new products. Because a CAMELS-like

100

115

120

125

convention for evaluating SWE models and estimation methods does not currently exist in the snow research and practice community, the core motivation for the work and the associated datasets is to provide a starting point in this direction, which would ideally yield CAMELS-like community benefits and impacts. The following sections present the approach and evaluation methods, describe the SWE datasets and results, and discuss the findings and offer conclusions and recommendations.

## 2 Approach

105 The overall objective of this paper is to propose and illustrate the use of a standard protocol for SWE product evaluation, using ASO SWE as a benchmark to evaluate the performance of the multiple SWE products. Supporting this goal, we obtain and process an extensive suite of distributed SWE product datasets, as well as the ASO SWE (reference dataset). These are spatially aggregated to a catchment scale (using the NOAA NextGen hydrofabric catchments dataset (median area ~7 km²; describe further below) to enable a spatially consistent inter-comparison. This section describes the datasets evaluated in this study, the processing methods, the evaluation metrics, and experimental protocol concept.

## 2.1 Study area

The evaluation was conducted across basins in the western United States (US) that have been surveyed by the Airborne Snow Observatory (ASO) from 2013–2024. These basins span much of the western US, including the Sierra Nevada (e.g., Tuolumne, Merced, San Joaquin, Kings), headwater basins in Colorado (e.g., Gunnison, Rio Grande, Conejos), as well as select basins in Utah, Oregon, Washington, and Wyoming. ASO has flown over 40 different unique basins covering more than 80,000 km². The basins range from maritime snow climates of California and the Pacific Northwest to more continental conditions in the Rockies, capturing a wide variety of snow accumulation and melt regimes. Flight frequency varies across basins, with the Tuolumne River Basin having the longest record (>60 flights over 12 years) and most other basins sampled 1–5 times per water year during accumulation and melt periods. ASO coverage is not continuous, as flight frequency depends on funding and client purchase. As a result, most basins have only a handful of years of record. Despite the limited temporal coverage, the diversity of basins with ASO data provides a robust basis for benchmarking SWE products. A summary of the ASO scenes (i.e., a single flight) used in this study by year and basin is given in **Table A.1**.

Upon evaluation of the ASO data, a small number of scenes were excluded from analysis due to incomplete coverage or known data quality issues. Specifically, three flights in the Kaweah River Basin during the 2019 season, one 2016 flight in the Rio Grande River Basin, and one 2014 flight in the Uncompahgre River Basin were removed — the latter due to erroneous geolocation metadata (latitude/longitude variables) that incorrectly positioned the scene in California rather than Colorado. The Kaweah and Rio Grande flights were excluded due to incomplete or missing SWE data within the basin. In addition, the January 29, 2017 flight covering the combined Cherry Eleanor and Tuolumne region (USCATE) was omitted due to missing data in the Cherry Eleanor Basin; separate January 29, 2017 flights for the individual Cherry Eleanor and

Tuolumne basins were retained instead. The two 2016 Olympic Peninsula flights were also excluded due to high vegetation-induced errors in this region (Cao et al., 2018). After removal of these flights, a total of 405 ASO flights remained for evaluation.

## 2.2 SWE products

The SWE datasets used in this procedure have varying designs and resolutions, as summarized in Table 1. These products collectively span three of the four categories of SWE estimates defined in the introduction (i.e., not including direct satellite imagery methods). Each category of SWE product has several product entries, which may yield insights into the strengths or weaknesses of a type of approach, as well as the individual products. Each of the datasets is described in more detail in the sub-sections following the table.

| SWE Dataset | Citation                    | Spatial<br>Resolution | Temporal<br>Coverage     | Туре | Methodology                                                          |
|-------------|-----------------------------|-----------------------|--------------------------|------|----------------------------------------------------------------------|
| ASO SWE     | Painter et al. (2016)       | 50 m                  | 04/03/2013 – present     | _    | SWE derived from 3m SD and modeled snow density.                     |
| SNODAS      | NOHRSC. 2004.               | 1 km                  | 9/30/2003 – present      | R    | Real-time, physics-based model with observational data assimilation. |
| SWANN-4km   | Broxton et al. (2019)       | 4 km                  | 10/1/1981 – present      | D    | Assimilates in-situ SWE/SD + 4km PRISM temp/precip.                  |
| SWANN-800m  | Broxton et al. (2024)       | 800 m                 | 10/1/1981 -<br>present   | D    | Assimilates in-situ SWE/SD + 800m PRISM temp/precip.                 |
| NWM SWE     | Cosgrove et al. (2024)      | 1 km                  | 2/1/1979 -<br>1/31/2023  | M    | NWM v3.0 forced with AORC v1.1                                       |
| CONUS404    | Rasmussen et al. (2023)     | 4 km                  | 10/1/1981 -<br>9/30/2021 | M    | WRF v3.9.1 with packages such as Noah-MP LSM.                        |
| UCLA SWE    | Margulis et al. (2019)      | 16 arc-sec (~500m)    | 10/1/1984 -<br>9/30/2021 | R    | SWE reanalysis assimilating Landsat/MODIS fSCA.                      |
| ERA5-Land   | Munoz-Sabater et al. (2021) | 9 km                  | 10/1/1950 -<br>present   | R    | Land reanalysis forced by ERA5 atmospheric fields.                   |



| SWEMLv2                | Johnson et al. (2024)  | 1 km                     | 1/1/2014 –<br>9/30/2023  | D | Assimilates SNOTEL and CDEC with lidar terrain features. |
|------------------------|------------------------|--------------------------|--------------------------|---|----------------------------------------------------------|
| NLDAS2-Noah<br>LSM     | Xia et al. (2012)      | 1/8 <sup>th</sup> degree | 1/2/1979 - present       | M | Noah LSM forced by NLDAS2 forcing data (53 fields).      |
| NLDAS2 –<br>Mosaic LSM | Xia et al. (2012)      | 1/8 <sup>th</sup> degree | 1/2/1979 - present       | M | Noah LSM forced by NLDAS2 forcing data (38 fields).      |
| NLDAS2 – VIC<br>LSM    | Xia et al. (2012)      | 1/8 <sup>th</sup> degree | 1/2/1979 - present       | M | VIC LSM forced by NLDAS2 forcing data (44 fields).       |
| Daymet                 | Thornton et al. (2021) | 1 km                     | 1/1/1980 –<br>12/31/2023 | M | Temp-based snow model using daily Tmin, Tmax, and Prcp.  |

Table 1: Summary of SWE product datasets used in this study (including the ASO reference observational estimates). Dataset type abbreviations: model (M), reanalysis/analysis (R), data-driven (D).

#### 2.2.1 Airborne Snow Observatory SWE

Since 2013, ASO has provided high-resolution (50 m) gridded estimates of snow water equivalent (SWE) based on lidar-derived snow depth combined with modeled snow density from the iSnobal energy balance snow model (Marks et al., 1999; Trujillo et al., 2025). Snow depth is measured from repeated airborne lidar scans of snow-on conditions, which are differenced against the snow-off (i.e., bare ground) conditions. The resulting SWE product has been shown to provide accurate spatially distributed SWE information (Painter et al., 2016) for discrete snapshots in time and individual watersheds. ASO SWE does contain uncertainties that arise both from errors in depth measurements and in density estimates. The ASO snow depth mean absolute error is reported to be 

from a numerical weather prediction (NWP) model, Rapid Update Cycle (RUC2). The model is updated (through data assimilation) by ground-based point data – Natural Resources Conservation Service (NRCS) SNOTEL SWE, California Department of Water Resources (CADWR) & BC Hydro SWE, and Cooperative Observer (COOP) SWE and depth – airborne (NOHRSC Airborne Gamma SWE), and satellite (NOHRSC Geostationary Operational Environmental Satellite [GOES] / Advanced Very High Resolution Radiometer [AVHRR] snow cover) data (Carroll et al., 2001). SNODAS provides additional snowpack variables such as snow depth and snow cover; however, only SWE was used in this study. The 1-km gridded product covers the continental US and extends from September 30, 2003, to the present.

## 2.2.3 UA/SWANN SWE






The Snow Water Artificial Neural Network (SWANN; Broxton et al., 2016; 2024) SWE dataset was developed at the University of Arizona. SWANN assimilates in-situ SWE and snow depth measurements from SNOTEL and National Weather Service's (NWS) COOP network stations as well as modeled, 4 km gridded temperature and precipitation data from PRISM. The SWE and snow depth were assimilated using a snow density model described in Dawson et al. (2017). SWE is normalized by accumulated snow (measured at SNOTEL stations) and modeled snow ablation, then interpolated between the station locations. The interpolated results are then used to correct gridded estimates of SWE generated from PRISM precipitation and temperature data. The 800 m SWANN SWE dataset uses 800 m PRISM precipitation and temperature data, whereas the daily 4 km PRISM data is downscaled to 800 m using the relationship between the 800 m and 4 km month PRISM climatologies (Daly et al., 2008). In addition to this assimilation framework, SWANN incorporates a machine learning component: artificial neural networks are trained to predict terrain- and vegetation-dependent adjustment factors, which scale the baseline SWE estimates to account for variations across different landscape positions (Broxton et al., 2024). This hybrid approach combines a physically based snow model, station assimilation, and ANN corrections. The 800 m and 4 km SWANN products are generated using the same process, and in general, the 800 m SWE dataset is similar to the 4 km dataset with minor differences at local scales (Broxton et al. 2024). Both datasets have a daily temporal resolution from October 1st, 1981, to September 30th, 2023.

# **2.2.4 UCLA-SWE**

This dataset is a snow reanalysis over the western US that assimilates fractional snow-covered area (fSCA) observations derived from the satellite-based Landsat and MODIS missions. It uses a Bayesian data assimilation framework, in which an ensemble snow energy balance model first produces estimates of a SWE prior distribution driven solely by meteorological forcings. These prior estimates are then updated by assimilating full-season (applied at the end of the snow period) fSCA observations to produce posterior estimates of SWE (Margulis et al., 2019). The resulting dataset contains SWE, fSCA, and snow depth. It has five ensemble statistics (mean, standard deviation, median, 25th and 75th percentiles) from fifty replicates (samples from the posterior distribution) for each snow estimate. The spatial coverage of the dataset is 31 N to 49 N and 125 W to 102 W. The dataset has also been successfully applied in the Andes and High Mountain Asia (Fang et al., 2022).




Verification of this dataset using ASO SWE over the Tuolumne River Basin has been conducted previously (Margulis et al., 2019), finding that spatial correlation was greater than 0.8 for WYs 2015-2017. The spatial resolution is 16 arc-seconds (~500 m), and a daily temporal resolution. The evaluation was done for WYs 2015 (dry), 2016 (average), and 2017 (wet) (Margulis et al., 2019). A more extensive evaluation of the UCLA-SWE product is conducted in this presented intercomparison.

#### **2.2.5 NWM SWE**

The NOAA National Water Model (NWM) retrospective dataset is produced by an operational US-wide hydrological modeling system that is based on the WRF-Hydro model (Gochis et al., 2020). SWE is simulated by the column land surface model component, the Noah Multi-Parameterization land surface model (Noah-MP; Cosgrove et al., 2024). The NWM has a multilayered, energy-balance-based snowpack formulation which tracks the snowpack condition across the entire NWM extent on a 1 km grid. NWM version 3.0 was used for this research, with a 44-year retrospective simulation (1979-2023). Previous evaluations of the NWM and the Noah-MP snow simulation have been conducted across a variety of sites in the Western United States (Garousi-Nejad & Tarboton, 2022; von Kaenel & Margulis, 2024).

## 205 **2.2.6 CONUS404**

CONUS404 (Rasmussen et al., 2023) is a coupled land-atmosphere simulation of the Weather Research and Forecasting (WRF; version 3.9.1) that was run by NSF National Center for Atmospheric Research (NCAR) in collaboration with the USGS. Boundary atmospheric conditions are derived from the European Centre for Medium-Range Weather Forecasting (ECMWF) atmospheric reanalysis of the global climate dataset (ERA5; Hersbach et al., 2020), which are provided on a 30 km grid. CONUS404 covers the contiguous US (CONUS) and parts of southern Canada and Northern Mexico for 40 years (water years 1980-2020) at a 4 km resolution. The WRF physics packages used were Thompson and Eidhammer (2014) microphysics, Yonsei University planetary boundary layer scheme, the Rapid Radiative Transfer Model (RRTMG) radiation scheme, and Noah-MP land surface model.

#### **2.2.7 ERA5-Land**

ERA5-Land is an operational global land reanalysis produced by ECMWF. It is based on the Carbon Hydrology-Tiled ECMWF Scheme for Surface Exchanges over Land (CHTESSEL) land surface model and is driven by atmospheric forcings from ERA5 meteorology. While ERA5-Land does not assimilate snow observations, ERA5 incorporates a comprehensive data assimilation system that influences its atmospheric variables. ERA5-Land provides over 50 land-surface variables, including snow depth and SWE, at hourly intervals and 9 km resolution (downscaled from ERA5's 31 km using interpolation). This resolution allows for improved representation of surface processes compared to earlier products such as ERA-Interim (~80 km). Evaluation studies show mixed results: ERA5-Land generally performs similarly for SWE to ERA5 overall, and with improved skill in snow-dominated regions above 1500 m, particularly in the Rockies (Muñoz-Sabater et al.,

2021). Given its global availability, relatively fine resolution for a reanalysis product, and frequent use in hydrologic studies, ERA5-Land provides a useful representative of global SWE products in this intercomparison.

## 225 **2.2.8 SWEML**



The Snow Water Equivalent Machine Learning version 2.0 (SWEMLv2.0) model is an open-source machine learning model trained to predict spatially continuous SWE in ASO basins throughout the Western U.S. (Johnson et al., 2024). Model training data include: static inputs of lidar-derived terrain attributes from the Copernicus GLO-90 DEM and Sturm regional snow classification; observational inputs including fractional snow-covered area derived from Visible Infrared Imaging Radiometry Suite (VIIRS; Riggs et al., 2017) normalized snow difference index (NSDI), neighbouring in-situ SWE observations from the SNOTEL station network, and gridded water-year-to-date precipitation from the North American Land Data Assimilation System (NLDAS). The model training target is ASO 50m resolution SWE. A user-selected spatial resolution (here, 1km) defines the spatial aggregation of gridded training data and model predictions. SWEMLv2.0 is an ML-based approach flexible in the ML algorithm selected, and in this case uses an extreme gradient-boosted decision tree method (XGBoost). A temporally independent model is trained for each year of record (2013–2023) by holding out all data from the prediction year from training and reserving 30% of the remaining data for hyperparameter optimization. For each held-out year of record, the respective trained model predicts gridded SWE over the study area.

# 2.2.9 NLDAS-2

NLDAS-2 is an operational data assimilation system featuring uncoupled land surface models which are driven by observation-based atmospheric forcing. NLDAS-2 uses three LSMs; Noah LSM (Ek et al., 2003), Mosaic (Koster & Suarez, 1996) and VIC (Variable Infiltration Capacity; Liang et al., 1994). Each LSM is operated at an hourly time step with a 1/8 degree spatial resolution (~12 km). The atmospheric forcings are derived from the North American Regional Reanalysis (NARR), precipitation from PRISM, and a ratio-based correction using GOES solar radiation data to bias-correct the NARR downward shortwave radiation. The Noah and Mosaic LSMs built on a surface-vegetation-atmosphere transfer (SVAT) framework focusing on water and energy exchanges between the land and atmosphere. The VIC LSM was developed within the hydrological community and, while accounting for a SVAT energy balance, is often used in research focusing on streamflow simulation. A notable feature of the VIC model implementation related to snow simulation is the inclusion of sub-grid elevation bands.

## 2.2.10

Daymet is a gridded surface meteorological dataset that provides daily estimates of key atmospheric variables across North America at a 1 km spatial resolution. The primary variables in Daymet include daily minimum temperature (Tmin), maximum temperature (Tmax), and precipitation (Prcp), which are estimated using interpolation and extrapolation from nearby weather stations. At each gridcell, weights are assigned for all surrounding stations within a search radius (Thornton



et al., 2021). Additionally, Daymet includes several secondary variables, including a simple estimate of SWE. This estimate of SWE is derived from the primary variables (Tmin, Tmax, Prcp) based on a simple temperature driven model of snow accumulation and melt (Thornton et al., 2000). Though Daymet SWE is simplified compared to physically based snow models, its high spatial resolution and national coverage make it a useful dataset for large-scale hydrological and snow studies.

## 2.3 Product standardization in the NextGen Hydrofabric

Because each of these SWE products differ in spatial resolution and extent, we must first implement a common spatial basis for comparison. Standardization was achieved using the NOAA NextGen Hydrofabric, a national geospatial dataset of connected rivers, lakes, and catchments designed for hydrologic modeling (Johnson, 2022). The hydrofabric is a refinement of the USGS NHDPlusV2 network, with short stream segments and small catchments removed to ensure hydrologically meaningful units, with catchments typically measuring 3–10 km² in area. This study uses Hydrofabric v2.2 (released October 2024), which provides catchments delineated by stream flowlines and topography, along with associated attributes such as elevation, slope, aspect, and land cover.

For each ASO basin, a subset of the hydrofabric was selected that excluded catchments not fully contained within the flight domain, ensuring that spatial aggregation was not biased by partially filled catchments. Gridded SWE datasets were spatially aggregated to the catchment resolution of each ASO hydrofabric by computing area-weighted overlaps between raster grid cells and hydrofabric polygons. For large-domain datasets, areally-conservative remapping was performed with the Python-based grid2poly tool and associated application scripts developed at NCAR, enabling computationally efficient processing. For small-domain, single-date analyses (e.g., ASO, SWEMLv2), the Python package xESMF was used to implement weighted averaging. Together, these approaches ensured consistent and reproducible aggregation of SWE across datasets. Figure 1 shows the remapping of ASO SWE in the Merced River Basin to the catchment resolution.

Figure 1: Illustration of the remapping of full-resolution ASO SWE in the Merced River Basin using the xESMF SpatialAverager tool. (left) Source gridded dataset; (middle) overlaid hydrofabric catchments (target polygons); (right) remapped hydrofabric catchment-average SWE.

# 280 2.4 SWE product evaluation metrics

Once remapped to the hydrofabric, SWE products were evaluated using relative mean absolute error (rMAE), percent bias (pbias), Kling-Gupta Efficiency (Gupta et al., 2009), and  $R^2$  (Equations 4.1 – 4.7). Metrics were calculated across all catchments in a scene, yielding one value per metric per scene.  $R^2$  was calculated as the square of the Pearson correlation coefficient between observed and predicted values.

$$MAE = \frac{1}{n} \sum_{i=1}^{n} |y_i - y|,$$
 (1)

$$rMAE = \frac{MAE}{mean(|y|)} \times 100 , \qquad (2)$$

$$PBIAS = 100 \times \frac{\sum_{i=1}^{n} (Y_{sim} - Y_{obs})}{\sum_{i=1}^{n} Y_{obs}},$$
(3)

$$KGE = 1 - \sqrt{(r-1)^2 + (a-1)^2 + (\beta-1)^2}, \ a = \frac{\sigma_{sim}}{\sigma_{obs}}, \ \beta = \frac{\mu_{sim}}{\mu_{obs}},$$
 (4-6)

$$R^2 = \left(\frac{Cov(O,P)}{\sigma_O \sigma_P}\right)^2,\tag{7}$$

## 290 2.5 Experimental protocol concept




This study follows a testbed framework that is being developed within the NOAA Cooperative Institute for Research to Operations in Hydrology (CIROH), termed the CIROH Hydrologic Prediction Testbed (CHPT). The term 'Testbed' has various formal and informal interpretations and implementations (e.g., Ralph et al., 2013). The CHPT comprises a collection of experimental protocols to facilitate quantitative community-wide benchmarking of different elements of the hydrologic prediction endeavor, including datasets, models and methods that are being developed in myriad research projects both within CIROH and more generally in the US and beyond. The CAMELS usage described in the Introduction is an example of an organic community protocol that emerged after the rapid adoption of CAMELS by the machine learning hydrology community (notably, Kratzert et al., 2019). The CHPT protocols include a number of defined elements, including the experiments to be conducted and/or datasets to be generated (e.g., basins, time periods), the verifying observations, the reference capability datasets, and the metrics for use in evaluating new innovations (Table 2).

Here, we focus on defining a protocol for catchment-based SWE spatial averages and discuss potential and challenges in specifying a common community SWE experiment. Ideally, an experimental protocol would: (1) use a subset of ASO SWE data for testing products in both "seen" (out of sample temporally) and "unseen" catchments (out of sample spatially); (2) harmonize temporal and spatial domains across diverse datasets, and (3) support future extension as new SWE products and ASO campaigns become available. We return to this objective in the Discussion section, after presenting the dataset analysis results.

| <b>Protocol Element</b> | Description                                                                                        |
|-------------------------|----------------------------------------------------------------------------------------------------|
| Objective               | The focus of the experiment, such as a particular capability for which the protocol seeks to       |
|                         | benchmark alternatives                                                                             |
| Observation(s)          | Datasets that can be used for validation and verification                                          |
| Reference capabilities  | An existing dataset, model version, system version, or method against which to assess the          |
| (baselines)             | marginal benefits of new innovations.                                                              |
| Experimental design     | The period, catchments, lead times if applicable, and other relevant details to ensure consistency |
|                         | in evaluating candidate innovations                                                                |
| Metrics of performance  | A range of common absolute and relative measures to evaluate performance across all                |
|                         | alternatives. Metrics may be split between 'core' and 'recommended' to prioritize focus on a       |
|                         | manageable metric set.                                                                             |
| Required innovation     | Information to identify significant elements used in each innovation (alternative) tested in a     |
| metadata                | protocol, including details such as model, method or dataset version, generation date, author,     |
|                         | among others.                                                                                      |
| Other considerations    | Qualitative considerations for capability evaluation (e.g., computational expense, portability,    |
|                         | complexity, dependencies, generalizability, potential operational latency).                        |
| References and related  | Key studies and relevant external activities.                                                      |
| activities              |                                                                                                    |
| Lead(s)/Contact(s)      | Active contributors or contact points, helping connect new participants and ensuring testbed       |
|                         | results are up-to-date.                                                                            |

Table 2: Elements of an experimental protocol for community benchmarking of geoscience results.

## 3 Results



This section shows intercomparison results across the large collection of ASO scenes at the catchment level defined by the NextGen hydrofabric, along with a regional perspective and conditioned on elevation. Because some products are at a coarser spatial resolution than the catchments, we also show analyses for basin-average SWE, which is relevant to applications such as predicting water supply (snowmelt runoff). Sample results for one of the proposed protocols are also presented. We note that the study's datasets and associated results provide several insights presented here, but these analyses are a subset of those potentially enabled through the standardization of analysis on a common SWE estimation protocol.

## 4.1 Catchment-level analyses

An initial overview of product performance at the catchment level (i.e., with one data point per catchment) across all basins and all ASO scenes is shown in Fig. 2. We find that UCLA-SWE and SWANN 800m performed similarly well with the

lowest rMAE values (29.8% and 31.8% respectively), high R<sup>2</sup> values (0.88 and 0.84 respectively) and KGE values (0.88 and 0.89 respectively), and the lowest percent bias (each 7.4%) other than ERA5-Land. Similarly, SNODAS and SWANN 4km performed comparably to each other and marginally worse than UCLA-SWE. Overall, this protocol found that the UCLA-SWE product consistently outperformed other SWE estimation methods, despite not assimilating common in-situ datasets (e.g. SNOTEL).

Figure 2: Product estimated SWE versus ASO SWE across all scenes from 2013 – 2024. NWM, UCLA-SWE, CONUS404, SWEMLv2, and Daymet do not include all ASO SWE data due to temporal limitations.



As shown in the inset statistics in Fig. 2, we calculated R<sup>2</sup>, percent bias, rMAE, and KGE for all ASO scenes (n > 400), with one value of each statistic for each scene. **Figure 3** summarizes these results as boxplots, highlighting both central tendency and variability in performance across basins. UCLA-SWE achieved the best median performance for rMAE, KGE, and R<sup>2</sup>, with a notably small spread, indicating consistent skill across regions. SWEMLv2 has a similarly small spread for KGE, rMAE, and R<sup>2</sup>. Performance for percent bias was more comparable across products, with SWANN (800 m, 4 km), UCLA-SWE, and SNODAS performing similarly, while ERA5-Land and the NWM had lower median biases, but a larger spread across the study basins. Other datasets involving versions of the Noah LSM model (in NLDAS2 and CONUS404) had a notable bias toward underestimation.

Figure 3: Boxplots of metric values for each SWE estimation product. Metrics values are calculated using all ASO and matching SWE product scenes across all regions. The boxes represent the interquartile range with a median line, while the whiskers represent the 10th and 90th percentiles.

Complementing the boxplots, **Fig. 4** presents the cumulative distribution of KGE, providing a probability-based view of model skill. This form of CDF-based statistical summary has become a core, widely-used analysis convention in the context of benchmarking streamflow simulation modeling approaches, as exemplified by use in summarizing results over the





CAMELS large-sample watershed dataset (Newman et al., 2017; Tang et al., 2025; Kratzert et al, 2024). UCLA-SWE, SWEMLv2 and SWANN products showed relatively higher performance, yielding median KGE values of 0.6 or above, with SNODAS also ranked competitively but slightly lower (0.57). The ERA5-Land and products involving NLDAS-NOAHLSM datasets exhibited the worst performance in this analysis, which may be due in part to their coarse spatial resolution. SWEMLv2 is unique amongst the products in having been trained on ASO scenes (notwithstanding temporal cross-validation).

Figure 4: Cumulative Distribution Function (CDF) of the Kling-Gupta Efficiency (KGE) of each SWE estimation product across all ASO study scenes (one KGE score per scene).

An attribute of SWE datasets that is of interest for both scientific research and societal applications is their ability to represent SWE across a range of elevations, which can be linked to the potential for predicting the elevation-dependent timing of snowmelt in river basins that are important for water supply, as well as to resilience or vulnerability to climate trends such as warming. **Figure 5** illustrates the performance of SWE products with elevation. We find that ERA5-Land generally overestimates SWE at low elevations and underestimates at high elevations, reflecting limitations of its coarse resolution. The NLDAS-2 LSMs consistently underpredict SWE, with a low bias increasing at higher elevations. Daymet underpredicts SWE at low—mid elevations and shows overestimation at high elevations in some California basins, a pattern absent in Colorado. In contrast, UCLA-SWE, SWEMLv2 and the SWANN products (4 km and 800 m) exhibit the lowest errors across all elevation bands, with SNODAS performing slightly worse than these products at higher elevations. This analysis across all of the study basins may obscure important regionally specific patterns. For example, focusing only on the Conejos River basin in Colorado, where elevations are generally higher, the same analysis shows that the products represent SWE well at low—mid elevations (<3400 m) but all products share a bias toward underpredicting at high elevations (>3400; Figure A.1), with the lowest biases emerging in the UCLA-SWE dataset.

Figure 5: Differences in predicted SWE relative to ASO SWE (modeled - ASO) across elevation bands for all ASO SWE scenes. The plot shows the density of points in SWE difference vs. elevation space, with color intensity indicating density of points (purple = high frequency, light blue = low frequency). Elevation bands are 0-2000m (low), 2000 - 2600m (low-mid), 2600 - 3200m (mid-high), >3200m (high). A locally weighted scatterplot smoothing regression (orange) highlights trends. MAE is reported for each elevation band (mm).



# 4.3 Regional analysis

Regional variability in product performance, as quantified by metrics that are sensitive to the climatological mean and spread of a variable (such as MAE and bias), can reflect differences that arise from differences in this snowpack climatology – e.g., contrasts between locations with deep, persistent snow versus shallower, intermittent snow. **Figure 6** repeats the analysis of Fig. 2, but separates Colorado from California scenes, which had the greatest number of scenes by region. Across the California basins, higher overall SWE leads to larger errors, whereas Colorado basins, with generally lower SWE, exhibit lower MAE but modestly reduced correlation (KGE) for most products. NLDAS-2 Noah LSM. SWEMLv2, and Daymet show better performance in Colorado. The reduced correlation and KGE in Colorado may be due to a range of factors, including lower quality of meteorological analyses (due to lower meteorological station density) to drive estimation methods and a greater variety of synoptic weather conditions driving snow accumulation, though this analysis is beyond the scope of this paper.

Figure 6: Product SWE versus ASO SWE for all California scenes (left) and all Colorado scenes (right). Note that the differing axes in the two regions: the maximum SWE for California is 3000 mm versus 2000 for Colorado. A linear regression line is plotted in red.




# 4.4 Basin-average SWE analyses

Because several of the SWE products are at a coarser spatial resolution on average than the hydrofabric catchments (e.g., NLDAS-2, ERA5-Land), they would not be expected to perform well in matching the spatial patterns of the hydrofabric-resolved scenes. We therefore also assess the performance of the products for basin-mean SWE, which would indicate the value of the SWE information at coarser scales, and one that is useful for applications such as basin-level snowmelt runoff prediction, e.g., water supply forecasting (e.g., Pagano et al., 2014). We compared basin-averaged SWE estimation performance using R², p-bias and rMAE across six basins with ≥20 ASO scenes. NLDAS-2 Noah LSM, a coarse-resolution product (1/8th degree), generally underperformed relative to other datasets, whereas the other coarse-resolution products such as ERA5-Land, NLDAS-2 Mosaic, and NLDAS-2 VIC LSM performed comparably to finer-resolution products at the basin scale (Fig. 7). Despite a small overestimation bias, SNODAS was also competitive (among the most skillful) in all of the basins evaluated in terms of R² and rMAE, an outcome that underscores its value for operational applications.

Figure 7: Basin-averaged SWE performance of different products compared to ASO SWE across six basins with ≥20 ASO scenes during 2013-2024, shown for (a) R², (b) percent bias, and (c) relative mean absolute error (rMAE). Each point represents a product's mean performance across all available scenes in a basin. The Tuolumne 2014–2021 subset (43 scenes) includes all products and is used for consistent intercomparison since not all products are available from 2013-2024.


In a final example of the potential application of this standardized approach for benchmarking a range of products, we evaluate the timing and magnitude of SWE among products, focusing on basin-averaged SWE in the Tuolumne River Basin (TRB), which had the greatest number of ASO scenes. Water years 2014 to 2016 were selected as a representative case (more for illustration than quantitative evaluation, due to the sample size) because all products were available and 27 ASO flights were conducted in the basin. **Figure 8** shows the basin-mean time series for each SWE product during that span. Across all SWE products, the NLDAS-2 Noah LSM exhibited notably early meltout compared to other datasets, while ERA5-Land tracked ASO most closely. SNODAS and SWANN produced the highest peak SWE, whereas Daymet showed the lowest peak but the most prolonged melt period. These patterns were consistent with broader multi-year comparisons (2011–2025, not shown), which generally indicated earlier melt in Noah LSM and higher peak SWE in SNODAS and SWANN.

Figure 8: Mean snow water equivalent in the Tuolumne River Basin for each dataset for water years 2014 - 2016. SWEMLv2 not included as it is not a timeseries.

# 5 Discussion



# 5.1 SWE dataset intercomparison

The broad intercomparison of different SWE products yielded a number of interesting insights. Notably, while the best performing datasets were the UCLA-SWE reanalysis product and the SWEMLv2 machine learning product, each of these had unique advantages over the others. UCLA-SWE benefited from advance knowledge of some inputs (such as seasonal melt-out date) while SWEMLv2 was the only product directly trained on the ASO scenes. Beyond these two, the best performing datasets according to different metrics included both process-based and data-driven techniques. The SWANN








and SWEML products represent the data-driven approach, and as they share methods, they represent the data-driven category narrowly. Notably, and perhaps surprisingly given its operational nature, the SNODAS product, a process modeling system that assimilates observations, rivaled the SWANN product in performance and outperformed the other purely physically-based modeling systems (such as the NWM or CONUS404 or NLDAS models) that do not assimilate observations. SNODAS is the closest product to a typical reanalysis that uses contemporaneous rather than future observational datasets to adjust its physical model states. While SWANN products tended to have lower bias, which is often expected for statistical products that are trained in a way that minimizes error, the SNODAS product had higher correlation skill, which can lead to lower overall squared error if the biases are not too large.

The UCLA-SWE product was the best performing product of those included in the framework, which shows that harnessing information retrospectively in a Bayesian statistical framework can prove beneficial for SWE estimation. As previously mentioned, the UCLA-SWE product includes adjustments based on Landsat and MODIS fractional snow covered area, though it does not assimilate in situ snow data (e.g., SNOTEL) like some of the other products. Due to its dependence on future information, it is infeasible as an operational product that can be used in prediction systems, but its overall quality gives it value as a benchmark dataset for evaluating other continuous space-time snow products or models in historical or retrospective evaluations.

We find that operational products are not notably worse than non-operational products, given that non-operational products such as CONUS404, UCLA-SWE, and ERA5-Land exhibited mixed or inferior performance to SNODAS. As noted earlier, the NLDAS products likely suffered in this evaluation due to their coarser scale, while the NWM's inclusion of the Noah-MP snow model, which has exhibited low snow biases in prior studies (e.g., Cho et al., 2022), may have also played a role. Of the three NLDAS-2 models, VIC had the highest correlation, perhaps a result of its sub-grid elevation representation, which enables higher snow accumulations and delayed melt at higher elevations, compared to a mean grid-cell elevation model. While performance varies across operational and non-operational products, strong results from UCLA-SWE and SNODAS show that operational products can approach retrospective reanalyses in quality.

A regional perspective separating California and Colorado indicated that California basins generally (across all products) exhibited higher correlation and efficiency metrics than Colorado basins, while Colorado basins had lower absolute errors, in part due to smaller SWE magnitudes. These differences likely reflect both hydroclimatic contrasts—such as higher SWE and more spatially coherent (frontal) storm systems in California versus lower SWE and more varied snow event generation patterns in Colorado—and the higher density of meteorological observations in California and associated quality of meteorological forcings for modeling. We speculate that these regional skill differences appear more tied to climatic and observational context than to fundamental model limitations. One feature of performance that appeared across nearly all products was a tendency for underestimation at high elevations (as illustrated in Fig. 5 and Fig. A.1), which could result from a number of factors, including a low bias at elevation in precipitation, lack of consideration of other atmospheric conditions such as wind speed and sublimation or snow redistribution, radiation exposure, or even biases in temperature lapse rates.





The comparison between the 4-km and 800-m SWANN products showed little difference in performance, suggesting that resolution improvements below the typical hydrofabric catchment scale (~7 km²) may not always benefit SWE estimation accuracy at that scale. Both products produced nearly identical skill metrics, with only marginal gains for the 800-m version. In contrast, products with coarser resolution (ERA5-Land, NLDAS-2 LSMs at ~10-12 km resolution) showed systematic biases, possibly linked to topographic smoothing. Their coarse grids often represented multiple subcatchments with a single value, leading to overestimation at low elevations and underestimation at high elevations. Nevertheless, some of these products (ERA5-Land, NLDAS-2 Mosaic and VIC) still reproduced basin-average SWE reasonably well (Fig. 7), as compensating catchment-scale errors tended to balance in aggregate.

We highlight again that ASO SWE is not a direct observation but rather a modeled estimate based on lidar-derived snow depth and modeled snow density (Painter et al., 2016). While ASO provides an invaluable reference for intercomparison, its uncertainties and limited spatial coverage mean that strong performance in ASO basins may not directly generalize to other regions.

### 5.2 Establishing community experimental protocols for SWE product evaluation and benchmarking

This paper has presented a suite of analyses for a range of available SWE products at a watershed scale that is relevant for hydrological modeling and applications. The broader objective of this paper, however, is to highlight the value of creating standardized experimental protocols for benchmarking SWE estimation capabilities. Such protocols reduce inconsistencies across research studies of basin selection, time period coverage, and validation strategies, creating a framework for direct comparison across datasets developed by different groups. Multiple challenges are present in defining a protocol that would be accepted widely across a community—such as discriminating products that incorporate ASO in training, aligning records with differing temporal spans, and balancing geographic diversity with sampling depth. Addressing these issues through community consensus can bring rigor and structure to benchmarking efforts, and lead to solid quantitative foundations for tracking progress in advancing our SWE estimation capabilities. We present an example of such a protocol and associated results, focusing on a geographic hydrofabric used as the spatial basis for the NOAA NextGen modeling system. The specification of the example protocol is also detailed in **Table A.2**.

### 480 5.2.1 Example of a protocol using NextGen Hydrofabric catchments

Choosing a testing suite requires balancing considerations of level of effort for researchers, adequate and representative sample sizes, and feasibility of holding out information for testing. A protocol could have a subset of 50 ASO SWE scenes from prominent basins (Tuolumne, Merced, Kings Canyon, Cherry Eleanor, and selected Colorado basins) over 3 years (2019–2021) that would be withheld from method training or development for testing. This set captures a representative range of snowpack conditions while leaving other scenes available for model training and assimilation. **Figure 9** shows the basins included. Certainly, alternative protocol/holdout strategies are possible, and we encourage the community to converge on one or more. It is likely to be difficult to strike a balance that allows all datasets contributed by researchers for evaluation

in the protocol to adhere completely to the protocol, but caveats to the comparisons can be noted along with their analysis results.

Figure 9: Example protocol set of prominent ASO basins across Colorado and California that could be held out for testing and standard evaluation.

# 5.3 Illustration of outcomes for the example protocol

To evaluate product performance within the proposed protocol, this analysis used ASO SWE data from prominent basins during WY2019–21. This protocol excludes extreme drought years (e.g., 2015 in the Sierras), which may bias evaluations toward wetter snow regimes, but it links to basins well represented in past literature (e.g., Tuolumne; Yang et al., 2023) and aligns with priority areas for future ASO campaigns (McCrindle, 2023). The protocol uses roughly 10% of all currently available ASO SWE data (**Table 3**), leaving most data available for assimilation or training of data-driven models.



| State      | Basin                    | Scenes | Subcatchments | Area (km2) |
|------------|--------------------------|--------|---------------|------------|
| California | Tuolumne                 | 5      | 145           | 1173       |
|            | Tuolumne Full            | 5      | 358           | 2778       |
|            | Kings Canyon - Southeast | 10     | 174           | 1225       |
|            | Merced                   | 4      | 111           | 827        |
|            | Merced Full              | 4      | 221           | 1710       |
|            | San Joaquin - Main Fork  | 10     | 172           | 1358       |
|            | Kaweah                   | 2      | 196           | 1438       |
| Colorado   | Blue River               | 4      | 100           | 866        |
|            | Gunnison East            | 2      | 95            | 749        |
|            | Gunnison - Taylor        | 2      | 89            | 658        |
|            | Conejos                  | 2      | 91            | 697        |
| Totals     | 11                       | 50     | 1752          | 13479      |

Table 3: List of ASO scenes and basin characteristics that are included in the WY2019 - WY2021 proposed protocol.

Boxplots of performance metric values (**Fig. 10**) show results consistent with the earlier all-scenes evaluation. UCLA-SWE maintained the highest R<sup>2</sup> and KGE and the lowest MAE, with percent bias closest to zero. Among data-driven models, SWANN 4km had the smallest median percent bias. Median KGE values largely mirrored the full dataset trends: **Figure 11** presents the CDF of KGE values, which is coarser than the all-scenes plot due to fewer observations. UCLA-SWE KGE (0.73) slightly increased, SWANN 4km decreased to 0.44, and NWM and CONUS404 improved by 0.14–0.16. Overall, UCLA-SWE remains the best benchmark product, while SNODAS is the strongest operational product for catchment SWE. SWEMLv2 is also highly competitive, with the caveat being its training on the basins in question (even though the testing years are out of sample).

Figure 10: Boxplots of metric values for each SWE estimation product. Metrics values are calculated for proposed basins between 2019 - 2021. The boxes represent the interquartile range with a median line, while the whiskers represent the 10th and 90th percentiles.

Figure 11: Cumulative Distribution Function (CDF) of the Kling-Gupta Efficiency (KGE) of each SWE estimation product and each according median KGE based on 2019 to 2021 ASO SWE in protocol basins.

#### 6. Conclusions





Estimating SWE at catchment scales represents a fundamental challenge in hydrology, and the growing diversity of SWE products being developed to address this challenge highlights the need for consistent evaluation frameworks. This study applied a standard approach for catchment-scale SWE evaluation across a wide range of available datasets covering the western US, using ASO as the reference observational dataset, and further outlined and demonstrated an example of an community evaluation experimental protocol drawing from the datasets used in this study.

Applying this framework to 12 SWE products revealed several useful insights, including: (1) snow reanalyses such as UCLA-SWE are currently the most accurate estimates of those reviewed; (2) some operational model-based products such as SNODAS remain competitive with newer machine learning approaches, while others (e.g., NWM) lag in quality; (3) data-driven methods tended to exhibit lower bias than physics-based models; (4) most of the products studied underestimated SWE at mid-to-high elevations; (5) products utilizing the Noah LSM and Noah-MP (e.g. NWM, CONUS404, NLDAS-2 Noah) tended to have undersimulation bias and relatively low performance; and (6) coarse resolution products (e.g., ERA5-Land, NLDAS-2 Mosaic, and NLDAS-2 VIC) showed skill at estimating SWE on a basin-wide scale despite struggling to validate as well at a catchment scale. Related prior studies such as Yang et al. (2023) similarly found that UCLA-SWE outperformed other products, though their evaluation was limited to fewer products and fewer ASO scenes. The present study expands on this work by including a wider array of datasets—spanning operational, reanalysis, and data-driven methods—and by leveraging a much larger reference dataset of ASO scenes. Within this broader benchmarking study, SNODAS emerged as more competitive than previously reported, even rivaling non-operational approaches, when viewed overall, though it performed less well in the example protocol.

These results yielded insights into the comparative accuracy of a wide range of current SWE products while illustrating the potential value of a standardized evaluation protocol. We described an example of an experimental protocol strategy that we hope may motivate further development and implementation of this concept by the snow science and applications community, including operational centers. Future protocol and study extensions could include grid- and point-based protocols, regional analyses as additional ASO data become available, and targeted assessments of model skill under different snow regimes, or protocols for other variables such as fSCA. With additional protocols, other snow observation sources (such as the SNOTEL network or satellite imagery) would be appropriate observational references. The example presented here focused on a hydrofabric catchment definition that is tailored to support the NOAA NextGen modeling effort, but a gridded spatial analysis basis (e.g., at 500m to 1km) may be of broader interest to the national and global community.

We recognize that motivating a community comprising non-coordinated research and development efforts to coalesce around shared experimentation faces hurdles, not the least of which is the need for any individual researcher to adopt external constraints and standards as outlined by a community protocol. Such shared experimental and analysis conventions have proven widely beneficial in galvanizing collaboration and progress in streamflow simulation, however, as is clearly evident in the widespread adoption and extension of the CAMELS dataset for model benchmarking. Such outcomes argue strongly

that the potential to obtain and track powerful, cross-community insights about a particular capability – in this case, catchment-level SWE estimation – makes pursuing such synergistic efforts worthwhile, both for individual researchers and community as a whole.

# **Appendices**


Figure A.1: SWE difference (modeled - ASO) versus elevation across all scenes in the Gunnison - Taylor River basin with a LOWESS regression (orange) to show trends. MAE calculated for each elevation level (0-3000m, 3000 - 3400m, +3400m

| State        | Basin                     | Scenes | Subcatchments | Area (km2)  | Temporal Coverage   |
|--------------|---------------------------|--------|---------------|-------------|---------------------|
|              | Tuolumne                  | 65     | 145           | 1173        | 20130403 - 20240506 |
|              | Tuolumne Full             | 20     | 358           | 2778        | 20200413 - 20240506 |
|              | Cherry Eleanor            | 36     | 64            | 450         | 20160401 - 20240506 |
|              | San Joaquin - South Fork  | 35     | 134           | 982         | 20170718 - 20240521 |
|              | Merced Full               | 20     | 221           | 1710        | 20200413 - 20240523 |
|              | Merced                    | 33     | 111           | 827         | 20200413 - 20240523 |
|              | Kings Canyon - South East | 27     | 174           | 1225        | 20150403 - 20240427 |
|              | Kings Canyon - Middle     | 26     | 109           | 824         | 20150404 - 20240427 |
|              | San Joaquin - Main Fork   | 25     | 172           | 1358        | 20170719 - 20240521 |
|              | SanJoaquin - Jose Willows | 26     | 263           | 1869        | 20180423 - 20240521 |
|              | Kings Canyon - North      | 23     | 98            | 648         | 20180426 - 20240427 |
|              | Lakes Basin               | 19     | 3             | 20          | 20150428 - 20190715 |
|              | Kings Canyon - South      | 19     | 52            | 410         | 20200411 - 20240427 |
|              | Kaweah                    | 15     | 196           | 1438        | 20210423 - 20240520 |
| California   | Yuba                      | 13     | 282           | 2133        | 20220325 - 20240527 |
| <b>G</b> aGa | Feather                   | 12     | 1100          | 8245        | 20220207 - 20240513 |
|              | Carson                    | 11     | 187           | 1366        | 20220311 - 20240516 |
|              | Truckee                   | 10     | 385           | 2789        | 20220310 - 20240517 |
|              | Rush Creek                | 8      | 23            | 121         | 20150326 - 20230702 |
|              | American                  | 7      | 432           | 3118        | 20230131 - 20240430 |
|              | Kern                      | 7      | 537           | 4054        | 20230204 - 20240508 |
|              | Lee Vining                | 4      | 17            | 100         | 20170717 - 20230702 |
|              | Upper Pit                 | 3      | 469           | 3750        | 20230208-20230513   |
|              | Lower Pit                 | 3      | 324           | 4417        | 20230200-20230513   |
|              | Middle Pit                | 3      | 413           | 3633        | 20230212 - 20230511 |
|              | Sacramento McCloud        | 3      | 444           | 3745        | 20230209 - 20230515 |
|              |                           | 3      | 38            | 290         |                     |
|              | Upper Owens               | 3      |               |             | 20230527 - 20230702 |
|              | Mono                      |        | 64            | 421         | 20230527 - 20230702 |
|              | Trinity                   | 1 11   | 247<br>100    | 1861<br>866 | 20240628            |
|              | Blue River                | 10     |               | 1111        | 20190419 - 20240605 |
|              | Conejos                   |        | 91            | 697         | 20150406 - 20240508 |
|              | Gunnison - Taylor         | 10     | 89            | 658         | 20180330 - 20240520 |
|              | Gunnison - East           | 10     | 95            | 749         | 20180331 - 20240520 |
|              | Dolores                   | 8      | 165           | 1292        | 20210420 - 20240430 |
|              | Windy Gap                 | 7      | 261           | 2042        | 20220418 - 20240530 |
|              | Roaring Fork              | 4      | 182           | 1722        | 20230411 - 20240522 |
|              | South Platte              | 4      | 155           | 1439        | 20230416 - 20240605 |
| Colorado     | Rio Grande River          | 2      | 361           | 3369        | 20150407 - 20150602 |
|              | Crested Butte             | 2      | 15            | 151         | 20160404 - 20180330 |
|              | Castle-Maroon             | 2      | 29            | 317         | 20190407 - 20190610 |
|              | Animas                    | 2      | 187           | 1814        | 20210419 - 20210515 |
|              | Big Thompson              | 2      | 121           | 986         | 20230521 - 20240421 |
|              | Boulder Creek             | 2      | 75            | 590         | 20230509 - 20240502 |
|              | Clear Creek               | 2      | 122           | 1017        | 20230509 - 20240502 |
|              | Poudre                    | 2      | 144           | 1026        | 20230522 - 20240415 |
|              | St Vrain Lefthand         | 2      | 78            | 684         | 20230521 - 20240421 |
|              | Yampa River               | 2      | 367           | 2708        | 20240411 - 20240527 |
| Oregon       | Sprague                   | 2      | 182           | 1469        | 20230514 - 20240314 |
| Utah         | Uinta                     | 3      | 240           | 1854        | 20240319 - 20240516 |
| Wyoming      | Green River               | 1      | 128           | 1038        | 20220611            |
| vvyoning     | Wind River                | 1      | 109           | 1052        | 20220611            |
| otals        | 51                        | 405    | 9951          | 80136       |                     |

Table A.1: ASO scenes used by Basin

| Protocol Element                                       | Description                                                                                                                                                                                                                                                                                                                                                                                             |  |  |  |
|--------------------------------------------------------|---------------------------------------------------------------------------------------------------------------------------------------------------------------------------------------------------------------------------------------------------------------------------------------------------------------------------------------------------------------------------------------------------------|--|--|--|
| Focus / Objective                                      | Daily mean areal SWE estimation at the NextGen HydroFabric catchment scale                                                                                                                                                                                                                                                                                                                              |  |  |  |
| Observation(s)                                         | Airborne Snow Observatory SWE zonally averaged to the Hydrofabric catchment scale                                                                                                                                                                                                                                                                                                                       |  |  |  |
| Reference capabilities                                 | NWM 3.0 SWE                                                                                                                                                                                                                                                                                                                                                                                             |  |  |  |
| (baselines)                                            | • SNODAS                                                                                                                                                                                                                                                                                                                                                                                                |  |  |  |
| Experimental design                                    | <ul> <li>Spatial Domain: Subset of ASO basins, e.g., a selection of watersheds across the western U.S., overlapping ASO flight lines and data availability.</li> <li>Spatial Resolution: Products are standardized in the NextGen hydrofabric.</li> <li>Study Period: 11 years of ASO data, temporally variable 2013-2024</li> <li>Timestep(s): daily SWE snapshots (according to ASO dates)</li> </ul> |  |  |  |
| Metrics of performance                                 | <ul> <li>Percent bias for quantifying over/underestimation of products</li> <li>KGE for assessing correlation and bias</li> <li>MAE to quantify the magnitude in error of SWE estimation</li> <li>NSE to evaluate products skill with respect to the observed mean</li> <li>Scores will also be expressed with respect to reference capability</li> </ul>                                               |  |  |  |
| Other factors                                          | <ul> <li>computational demand of innovation</li> <li>stability of dependencies (method, dataset)</li> <li>extent of existing documentation and published research</li> <li>dependency on proprietary resources or methods</li> </ul>                                                                                                                                                                    |  |  |  |
| Key protocol references and related activities (links) | <ul><li>Painter et al., 2016</li><li>Yang et al., 2023</li></ul>                                                                                                                                                                                                                                                                                                                                        |  |  |  |
| Innovation Metadata                                    | <ul> <li>product label</li> <li>model or dataset version</li> <li>generation date</li> <li>author</li> <li>institution</li> <li>product reference and/or link</li> </ul>                                                                                                                                                                                                                                |  |  |  |
| Current Contact(s)                                     | Ethan Ritchie, Andy Wood                                                                                                                                                                                                                                                                                                                                                                                |  |  |  |

Table A.2: Illustration of an example SWE experimental protocol for community benchmarking of catchment-level SWE product results.

| NASA ASO ASO <a href="https://nsidc.org/data/aso_50m_swe/versions/1">https://nsidc.org/data/aso_50m_swe/versions/1</a> SWE ASO Inc. <a href="https://data.airbornesnowobservatories.com/">https://data.airbornesnowobservatories.com/</a> SNODAS <a href="https://nsidc.org/data/g02158/versions/1">https://nsidc.org/data/g02158/versions/1</a> UA/SWANN <a href="https://climate.arizona.edu/data/UA_SWE/Daily">https://climate.arizona.edu/data/UA_SWE/Daily</a> NWM SWE <a href="https://climate.arizona.edu/data/UA_SWE/Daily">https://climate.arizona.edu/data/UA_SWE/Daily</a> NWM SWE <a href="https://registry.opendata.aws/nwm-archive/">https://registry.opendata.aws/nwm-archive/</a> UCLA SWE <a href="https://nsidc.org/data/wus_ucla_sr/versions/1">https://nsidc.org/data/wus_ucla_sr/versions/1</a> ERA5-Land <a "="" data.airbornesnowobservatories.com="" href="https://cds.climate.copernicus.eu/datasets/reanal-https://cds.climate.copernicus.eu/datasets/reanal-https://www.sciencebase.gov/catalog/item/63726&lt;/a&gt;&lt;/th&gt;&lt;th&gt;Nov 19, 2024&lt;/th&gt;&lt;/tr&gt;&lt;tr&gt;&lt;td&gt;SWE ASO Inc. &lt;a href=" https:="">https://data.airbornesnowobservatories.com/</a> SNODAS <a href="https://nsidc.org/data/g02158/versions/1">https://nsidc.org/data/g02158/versions/1</a> UA/SWANN <a href="https://climate.arizona.edu/data/UA_SWE/Daily">https://climate.arizona.edu/data/UA_SWE/Daily</a> NWM SWE <a href="https://registry.opendata.aws/nwm-archive/">https://climate.arizona.edu/data/UA_SWE/Daily</a> NWM SWE <a href="https://registry.opendata.aws/nwm-archive/">https://registry.opendata.aws/nwm-archive/</a> UCLA SWE <a href="https://nsidc.org/data/wus_ucla_sr/versions/1">https://nsidc.org/data/wus_ucla_sr/versions/1</a> ERA5-Land <a "="" data.airbornesnowobservatories.com="" href="https://cds.climate.copernicus.eu/datasets/reanal-nttps://cds.climate.copernicus.eu/datasets/reanal-nttps://cds.climate.copernicus.eu/datasets/reanal-nttps://cds.climate.copernicus.eu/datasets/reanal-nttps://cds.climate.copernicus.eu/datasets/reanal-nttps://cds.climate.copernicus.eu/datasets/reanal-nttps://cds.climate.copernicus.eu/datasets/reanal-nttps://cds.climate.copernicus.eu/datasets/reanal-nttps://cds.climate.copernicus.eu/datasets/reanal-nttps://cds.climate.copernicus.eu/datasets/reanal-nttps://cds.climate.copernicus.eu/datasets/reanal-nttps://cds.climate.copernicus.eu/datasets/reanal-nttps://cds.climate.copernicus.eu/datasets/reanal-nttps://cds.climate.copernicus.eu/datasets/reanal-nttps://cds.climate.copernicus.eu/datasets/reanal-nttps://cds.climate.copernicus.eu/datasets/reanal-nttps://cds.climate.copernicus.eu/datasets/reanal-nttps://cds.climate.copernicus.eu/datasets/reanal-nttps://cds.climate.copernicus.eu/datasets/reanal-nttps://cds.climate.copernicus.eu/datasets/reanal-nttps://cds.climate.copernicus.eu/datasets/reanal-nttps://cds.climate.copernicus.eu/datasets/reanal-nttps://cds.climate.copernicus.eu/datasets/reanal-nttps://cds.climate.copernicus.eu/datasets/reanal-nttps://cds.climate.copernicus.eu/datasets/reanal-nttps://cds.climate.copernicus.eu/datasets/reanal-nttps://cds.climate.copernicus.eu/datasets/reanal-nttps://c&lt;/td&gt;&lt;td&gt;Nov 19, 2024&lt;/td&gt;&lt;/tr&gt;&lt;tr&gt;&lt;td&gt;Inc. &lt;a href=" https:="">https://data.airbornesnowobservatories.com/</a> SNODAS <a href="https://nsidc.org/data/g02158/versions/1">https://nsidc.org/data/g02158/versions/1</a> UA/SWANN <a href="https://climate.arizona.edu/data/UA_SWE/Daily">https://climate.arizona.edu/data/UA_SWE/Daily</a> NWM SWE <a href="https://registry.opendata.aws/nwm-archive/">https://climate.arizona.edu/data/UA_SWE/Daily</a> NWM SWE <a href="https://registry.opendata.aws/nwm-archive/">https://registry.opendata.aws/nwm-archive/</a> UCLA SWE <a href="https://nsidc.org/data/wus_ucla_sr/versions/1">https://nsidc.org/data/wus_ucla_sr/versions/1</a> ERA5-Land <a href="https://cds.climate.copernicus.eu/datasets/reanal/">https://cds.climate.copernicus.eu/datasets/reanal/</a> <td></td> |                                                |
|-------------------------------------------------------------------------------------------------------------------------------------------------------------------------------------------------------------------------------------------------------------------------------------------------------------------------------------------------------------------------------------------------------------------------------------------------------------------------------------------------------------------------------------------------------------------------------------------------------------------------------------------------------------------------------------------------------------------------------------------------------------------------------------------------------------------------------------------------------------------------------------------------------------------------------------------------------------------------------------------------------------------------------------------------------------------------------------------------------------------------------------------------------------------------------------------------------------------------------------------------------------------------------------------------------------------------------------------------------------------------------------------------------------------------------------------------------------------------------------------------------------------------------------------------------------------------------------------------------------------------------------------------------------------------------------------------------------------------------------------------------------------------------------------------------------------------------------------------------------------------------------------------------------------------------------------------------------------------------------------------------------------------------------------------------------------------------------------------------------------------------------------------------------------------------------------------------------------------------------------------------------------------------------------------------------------------------------------------------------------------------------------------------------------------------------------------------------------------------------------------------------------------------------------------------------------------------------------------------------------------------------------------------------------------------------------------------------------------------------------------------------------------------------------------------------------------------------------------------------------------------------------------------------------------------------------------------------------------------------------------------------------------------------------------------------------------------------------------------------------------------------------------------------------------------------------------------------------------------------------------------------------------------------------------------------------------------------------------------------------------------------------------------------------------------------------------------------------------------------------------------------------------------------------------------------------------------------------------------------------------------------------------------------------------------------------------------------------------------------------------------------------------------------------------------------------------------------------------------------------------------------------------------------------------------------------------------------------------------------------------------------------------------------------------------------------------------------------------------------------------------------------|------------------------------------------------|
| SNODAS https://nsidc.org/data/g02158/versions/1  UA/SWANN https://climate.arizona.edu/data/UA_SWE/Daily  UA/SWANN https://climate.arizona.edu/data/UA_SWE/Daily  NWM SWE https://registry.opendata.aws/nwm-archive/  UCLA SWE https://nsidc.org/data/wus_ucla_sr/versions/1  ERA5-Land https://cds.climate.copernicus.eu/datasets/reanal/                                                                                                                                                                                                                                                                                                                                                                                                                                                                                                                                                                                                                                                                                                                                                                                                                                                                                                                                                                                                                                                                                                                                                                                                                                                                                                                                                                                                                                                                                                                                                                                                                                                                                                                                                                                                                                                                                                                                                                                                                                                                                                                                                                                                                                                                                                                                                                                                                                                                                                                                                                                                                                                                                                                                                                                                                                                                                                                                                                                                                                                                                                                                                                                                                                                                                                                                                                                                                                                                                                                                                                                                                                                                                                                                                                                                 |                                                |
| UA/SWANN  https://climate.arizona.edu/data/UA_SWE/Daily  UA/SWANN  https://climate.arizona.edu/data/UA_SWE/Daily  NWM SWE  https://registry.opendata.aws/nwm-archive/  UCLA SWE  https://nsidc.org/data/wus_ucla_sr/versions/1  ERA5-Land  https://cds.climate.copernicus.eu/datasets/reanal/                                                                                                                                                                                                                                                                                                                                                                                                                                                                                                                                                                                                                                                                                                                                                                                                                                                                                                                                                                                                                                                                                                                                                                                                                                                                                                                                                                                                                                                                                                                                                                                                                                                                                                                                                                                                                                                                                                                                                                                                                                                                                                                                                                                                                                                                                                                                                                                                                                                                                                                                                                                                                                                                                                                                                                                                                                                                                                                                                                                                                                                                                                                                                                                                                                                                                                                                                                                                                                                                                                                                                                                                                                                                                                                                                                                                                                             | July 2, 2025                                   |
| UA/SWANN  https://climate.arizona.edu/data/UA_SWE/Daily  NWM SWE https://registry.opendata.aws/nwm-archive/  UCLA SWE https://nsidc.org/data/wus_ucla_sr/versions/1  ERA5-Land https://cds.climate.copernicus.eu/datasets/reanal/                                                                                                                                                                                                                                                                                                                                                                                                                                                                                                                                                                                                                                                                                                                                                                                                                                                                                                                                                                                                                                                                                                                                                                                                                                                                                                                                                                                                                                                                                                                                                                                                                                                                                                                                                                                                                                                                                                                                                                                                                                                                                                                                                                                                                                                                                                                                                                                                                                                                                                                                                                                                                                                                                                                                                                                                                                                                                                                                                                                                                                                                                                                                                                                                                                                                                                                                                                                                                                                                                                                                                                                                                                                                                                                                                                                                                                                                                                         | July 3, 2025                                   |
| https://climate.arizona.edu/data/UA_SWE/Daily  NWM SWE https://registry.opendata.aws/nwm-archive/  UCLA SWE https://nsidc.org/data/wus_ucla_sr/versions/1  ERA5-Land https://cds.climate.copernicus.eu/datasets/reanal/                                                                                                                                                                                                                                                                                                                                                                                                                                                                                                                                                                                                                                                                                                                                                                                                                                                                                                                                                                                                                                                                                                                                                                                                                                                                                                                                                                                                                                                                                                                                                                                                                                                                                                                                                                                                                                                                                                                                                                                                                                                                                                                                                                                                                                                                                                                                                                                                                                                                                                                                                                                                                                                                                                                                                                                                                                                                                                                                                                                                                                                                                                                                                                                                                                                                                                                                                                                                                                                                                                                                                                                                                                                                                                                                                                                                                                                                                                                   | Data_800m/ May 15, 2025                        |
| NWM SWE <a href="https://registry.opendata.aws/nwm-archive/">https://registry.opendata.aws/nwm-archive/</a> UCLA SWE <a href="https://nsidc.org/data/wus_ucla_sr/versions/1">https://nsidc.org/data/wus_ucla_sr/versions/1</a> ERA5-Land <a href="https://cds.climate.copernicus.eu/datasets/reanal/">https://cds.climate.copernicus.eu/datasets/reanal/</a>                                                                                                                                                                                                                                                                                                                                                                                                                                                                                                                                                                                                                                                                                                                                                                                                                                                                                                                                                                                                                                                                                                                                                                                                                                                                                                                                                                                                                                                                                                                                                                                                                                                                                                                                                                                                                                                                                                                                                                                                                                                                                                                                                                                                                                                                                                                                                                                                                                                                                                                                                                                                                                                                                                                                                                                                                                                                                                                                                                                                                                                                                                                                                                                                                                                                                                                                                                                                                                                                                                                                                                                                                                                                                                                                                                              |                                                |
| UCLA SWE <a href="https://nsidc.org/data/wus_ucla_sr/versions/1">https://nsidc.org/data/wus_ucla_sr/versions/1</a> ERA5-Land <a cds.climate.copernicus.eu="" datasets="" href="https://cds.climate.copernicus.eu/datasets/reanal-nttps://cds.climate.copernicus.eu/datasets/reanal-nttps://cds.climate.copernicus.eu/datasets/reanal-nttps://cds.climate.copernicus.eu/datasets/reanal-nttps://cds.climate.copernicus.eu/datasets/reanal-nttps://cds.climate.copernicus.eu/datasets/reanal-nttps://cds.climate.copernicus.eu/datasets/reanal-nttps://cds.climate.copernicus.eu/datasets/reanal-nttps://cds.climate.copernicus.eu/datasets/reanal-nttps://cds.climate.copernicus.eu/datasets/reanal-nttps://cds.climate.copernicus.eu/datasets/reanal-nttps://cds.climate.copernicus.eu/datasets/reanal-nttps://cds.climate.copernicus.eu/datasets/reanal-nttps://cds.climate.copernicus.eu/datasets/reanal-nttps://cds.climate.copernicus.eu/datasets/reanal-nttps://cds.climate.copernicus.eu/datasets/reanal-nttps://cds.climate.copernicus.eu/datasets/reanal-nttps://cds.climate.copernicus.eu/datasets/reanal-nttps://cds.climate.copernicus.eu/datasets/reanal-nttps://cds.climate.copernicus.eu/datasets/reanal-nttps://cds.climate.copernicus.eu/datasets/reanal-nttps://cds.climate.copernicus.eu/datasets/reanal-nttps://cds.climate.copernicus.eu/datasets/reanal-nttps://cds.climate.copernicus.eu/datasets/reanal-nttps://cds.climate.copernicus.eu/datasets/reanal-nttps://cds.climate.copernicus.eu/datasets/reanal-nttps://cds.climate.copernicus.eu/datasets/reanal-nttps://cds.climate.copernicus.eu/datasets/reanal-nttps://cds.climate.copernicus.eu/datasets/reanal-nttps://cds.climate.copernicus.eu/datasets/reanal-nttps://cds.climate.copernicus.eu/datasets/reanal-nttps://cds.climate.copernicus.eu/datasets/reanal-nttps://cds.climate.copernicus.eu/datasets/reanal-nttps://cds.climate.copernicus.eu/datasets/reanal-nttps://cds.climate.copernicus.eu/datasets/reanal-nttps://cds.climate.copernicus.eu/datasets/reanal-nttps://cds.climate.copernicus.eu/datasets/reanal-nttps://cds.climate.copernicus.eu/datasets/reanal-nttps://cds.&lt;/td&gt;&lt;td&gt;&lt;u&gt;Data_4km/&lt;/u&gt; May 15, 2025&lt;/td&gt;&lt;/tr&gt;&lt;tr&gt;&lt;td&gt;ERA5-Land &lt;a href=" https:="" reanal-nata-2-2-2-2-2-2-2-2-2-2-2-2-2-2-2-2-2-2-<="" td=""><td>Jan 22, 2025</td></a>                                                                                                                                                                                                                                                                                                                                                                                                                                                                                                                                                                                                                                                                                                                                                                                                                                                                                                                                                                                                                                                                                                                                                                                                                                                                                                                                                                                                                                                                                                                                                                                                                                                                                                                                      | Jan 22, 2025                                   |
| https://cds.climate.copernicus.eu/datasets/reanal                                                                                                                                                                                                                                                                                                                                                                                                                                                                                                                                                                                                                                                                                                                                                                                                                                                                                                                                                                                                                                                                                                                                                                                                                                                                                                                                                                                                                                                                                                                                                                                                                                                                                                                                                                                                                                                                                                                                                                                                                                                                                                                                                                                                                                                                                                                                                                                                                                                                                                                                                                                                                                                                                                                                                                                                                                                                                                                                                                                                                                                                                                                                                                                                                                                                                                                                                                                                                                                                                                                                                                                                                                                                                                                                                                                                                                                                                                                                                                                                                                                                                         | Jan 23, 2025                                   |
| CONUS404 https://www.sciencebase.gov/catalog/item/63720                                                                                                                                                                                                                                                                                                                                                                                                                                                                                                                                                                                                                                                                                                                                                                                                                                                                                                                                                                                                                                                                                                                                                                                                                                                                                                                                                                                                                                                                                                                                                                                                                                                                                                                                                                                                                                                                                                                                                                                                                                                                                                                                                                                                                                                                                                                                                                                                                                                                                                                                                                                                                                                                                                                                                                                                                                                                                                                                                                                                                                                                                                                                                                                                                                                                                                                                                                                                                                                                                                                                                                                                                                                                                                                                                                                                                                                                                                                                                                                                                                                                                   | ysis-era5-land?tab=download July 3, 2025       |
|                                                                                                                                                                                                                                                                                                                                                                                                                                                                                                                                                                                                                                                                                                                                                                                                                                                                                                                                                                                                                                                                                                                                                                                                                                                                                                                                                                                                                                                                                                                                                                                                                                                                                                                                                                                                                                                                                                                                                                                                                                                                                                                                                                                                                                                                                                                                                                                                                                                                                                                                                                                                                                                                                                                                                                                                                                                                                                                                                                                                                                                                                                                                                                                                                                                                                                                                                                                                                                                                                                                                                                                                                                                                                                                                                                                                                                                                                                                                                                                                                                                                                                                                           | <u>cd09d34ed907bf6c6ab1</u> Jan 2025           |
| NLDAS2- https://disc.gsfc.nasa.gov/datasets/NLDAS_NO_                                                                                                                                                                                                                                                                                                                                                                                                                                                                                                                                                                                                                                                                                                                                                                                                                                                                                                                                                                                                                                                                                                                                                                                                                                                                                                                                                                                                                                                                                                                                                                                                                                                                                                                                                                                                                                                                                                                                                                                                                                                                                                                                                                                                                                                                                                                                                                                                                                                                                                                                                                                                                                                                                                                                                                                                                                                                                                                                                                                                                                                                                                                                                                                                                                                                                                                                                                                                                                                                                                                                                                                                                                                                                                                                                                                                                                                                                                                                                                                                                                                                                     | AH0125 H_2.0/summary?keywords=N                |
| Noah LSM <u>LDAS</u>                                                                                                                                                                                                                                                                                                                                                                                                                                                                                                                                                                                                                                                                                                                                                                                                                                                                                                                                                                                                                                                                                                                                                                                                                                                                                                                                                                                                                                                                                                                                                                                                                                                                                                                                                                                                                                                                                                                                                                                                                                                                                                                                                                                                                                                                                                                                                                                                                                                                                                                                                                                                                                                                                                                                                                                                                                                                                                                                                                                                                                                                                                                                                                                                                                                                                                                                                                                                                                                                                                                                                                                                                                                                                                                                                                                                                                                                                                                                                                                                                                                                                                                      | July 3, 2025                                   |
| NLDAS2 – <a href="https://disc.gsfc.nasa.gov/datasets/NLDAS_MO">https://disc.gsfc.nasa.gov/datasets/NLDAS_MO</a>                                                                                                                                                                                                                                                                                                                                                                                                                                                                                                                                                                                                                                                                                                                                                                                                                                                                                                                                                                                                                                                                                                                                                                                                                                                                                                                                                                                                                                                                                                                                                                                                                                                                                                                                                                                                                                                                                                                                                                                                                                                                                                                                                                                                                                                                                                                                                                                                                                                                                                                                                                                                                                                                                                                                                                                                                                                                                                                                                                                                                                                                                                                                                                                                                                                                                                                                                                                                                                                                                                                                                                                                                                                                                                                                                                                                                                                                                                                                                                                                                          | S0125_H_2.0/summary?keywords=NL                |
| Mosaic LSM <u>DAS</u>                                                                                                                                                                                                                                                                                                                                                                                                                                                                                                                                                                                                                                                                                                                                                                                                                                                                                                                                                                                                                                                                                                                                                                                                                                                                                                                                                                                                                                                                                                                                                                                                                                                                                                                                                                                                                                                                                                                                                                                                                                                                                                                                                                                                                                                                                                                                                                                                                                                                                                                                                                                                                                                                                                                                                                                                                                                                                                                                                                                                                                                                                                                                                                                                                                                                                                                                                                                                                                                                                                                                                                                                                                                                                                                                                                                                                                                                                                                                                                                                                                                                                                                     | July 3, 2025,                                  |
| NLDAS2 – <a href="https://disc.gsfc.nasa.gov/datasets/NLDAS_VIC">https://disc.gsfc.nasa.gov/datasets/NLDAS_VIC</a>                                                                                                                                                                                                                                                                                                                                                                                                                                                                                                                                                                                                                                                                                                                                                                                                                                                                                                                                                                                                                                                                                                                                                                                                                                                                                                                                                                                                                                                                                                                                                                                                                                                                                                                                                                                                                                                                                                                                                                                                                                                                                                                                                                                                                                                                                                                                                                                                                                                                                                                                                                                                                                                                                                                                                                                                                                                                                                                                                                                                                                                                                                                                                                                                                                                                                                                                                                                                                                                                                                                                                                                                                                                                                                                                                                                                                                                                                                                                                                                                                        | 0125_H_2.0/summary?keywords=NLD                |
| VIC LSM <u>AS</u>                                                                                                                                                                                                                                                                                                                                                                                                                                                                                                                                                                                                                                                                                                                                                                                                                                                                                                                                                                                                                                                                                                                                                                                                                                                                                                                                                                                                                                                                                                                                                                                                                                                                                                                                                                                                                                                                                                                                                                                                                                                                                                                                                                                                                                                                                                                                                                                                                                                                                                                                                                                                                                                                                                                                                                                                                                                                                                                                                                                                                                                                                                                                                                                                                                                                                                                                                                                                                                                                                                                                                                                                                                                                                                                                                                                                                                                                                                                                                                                                                                                                                                                         | July 3, 2025                                   |
| Daymet <a href="https://www.earthdata.nasa.gov/data/catalog/orn">https://www.earthdata.nasa.gov/data/catalog/orn</a>                                                                                                                                                                                                                                                                                                                                                                                                                                                                                                                                                                                                                                                                                                                                                                                                                                                                                                                                                                                                                                                                                                                                                                                                                                                                                                                                                                                                                                                                                                                                                                                                                                                                                                                                                                                                                                                                                                                                                                                                                                                                                                                                                                                                                                                                                                                                                                                                                                                                                                                                                                                                                                                                                                                                                                                                                                                                                                                                                                                                                                                                                                                                                                                                                                                                                                                                                                                                                                                                                                                                                                                                                                                                                                                                                                                                                                                                                                                                                                                                                      | l-cloud-daymet-daily-v4r1-2129-4.1 Aug 3, 2025 |
| SWEMLv2                                                                                                                                                                                                                                                                                                                                                                                                                                                                                                                                                                                                                                                                                                                                                                                                                                                                                                                                                                                                                                                                                                                                                                                                                                                                                                                                                                                                                                                                                                                                                                                                                                                                                                                                                                                                                                                                                                                                                                                                                                                                                                                                                                                                                                                                                                                                                                                                                                                                                                                                                                                                                                                                                                                                                                                                                                                                                                                                                                                                                                                                                                                                                                                                                                                                                                                                                                                                                                                                                                                                                                                                                                                                                                                                                                                                                                                                                                                                                                                                                                                                                                                                   | Provided by the                                |
| Johnson et al., 2024                                                                                                                                                                                                                                                                                                                                                                                                                                                                                                                                                                                                                                                                                                                                                                                                                                                                                                                                                                                                                                                                                                                                                                                                                                                                                                                                                                                                                                                                                                                                                                                                                                                                                                                                                                                                                                                                                                                                                                                                                                                                                                                                                                                                                                                                                                                                                                                                                                                                                                                                                                                                                                                                                                                                                                                                                                                                                                                                                                                                                                                                                                                                                                                                                                                                                                                                                                                                                                                                                                                                                                                                                                                                                                                                                                                                                                                                                                                                                                                                                                                                                                                      | Author                                         |

Table A.3: SWE Product sources and access dates

SNODAS data is available at the NSIDC at https://nsidc.org/data/g02158/versions/1 as of February 23, 2025.

# Data availability

Processed datasets will be made available at the hydrofabric resolution for each basin in an online resource with a DOI, once the paper is accepted for publication. This statement will be updated accordingly.

https://doi.org/10.5194/egusphere-2025-5514 Preprint. Discussion started: 24 November 2025

© Author(s) 2025. CC BY 4.0 License.

NASA/JPL-era ASO data (2013–2019) are publicly available via NSIDC at <a href="https://nsidc.org/data/aso\_50m\_swe/versions/1">https://nsidc.org/data/aso\_50m\_swe/versions/1</a>, while more recent (2020 - present) ASO datasets and flight reports are accessible through ASO Inc. at <a href="https://www.airbornesnowobservatories.com/">https://www.airbornesnowobservatories.com/</a> as of August 15th, 2025.

Other SWE product sources are listed in **Table A.3**.

## Acknowledgments

This research was supported by NOAA Cooperative Institute for Research to Operations in Hydrology (CIROH) grants to the Colorado School of Mines (CSM) and the University of Alabama (UA), with funding under award NA22NWS4320003 from the NOAA Cooperative Institute Program. The statements, findings, conclusions, and recommendations are those of the author(s) and do not necessarily reflect the opinions of NOAA or the author hosting institutions. We also acknowledge advice from Dr. Jeff Deems of ASO, Inc. in initially obtaining ASO SWE datasets, and for comments on the permissibility of the ASO dataset usage. We thank Nicholas Danes from CSM's research computing staff for support. The protocol strategy described here is a core element of the CIROH Hydrologic Prediction Testbed (CHPT).

AI (ChatGPT) was used in developing an early draft of this manuscript. AI was solely used for editing and restructuring text for clarity and conciseness. The authors provided all scientific content, analysis, and interpretation. All AI-generated material was reviewed and revised for accuracy in later drafts. The authors take full responsibility for the content of the presented paper.

#### **Author contribution**



ER performed all data processing and analyses appearing in the plots and tables of the paper, and wrote the first draft. AW led the overall study and the testbed-based benchmarking project at CSM, designed the protocol strategy, supplied several of the datasets and the methods used in the work, and co-wrote the final draft. AM and RJ provided review and input helping to shape the analyses, while RJ led a connected SWE data assimilation project and SWEML2 development at UA. JS provided technical support on using remapping methods. DL and EG supplied the SWEML2 datasets and description. All co-authors provided comments and/or proofing on the final draft.

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
