# Peer review of "Benchmarking Catchment-Scale Snow Water Equivalent Datasets and Models in the Western United States"

_EGUsphere, 2025_

## Referee Comment (RC2)

**January 20th, 2026**
Reviewer: Jack Tarricone

Benchmarking Catchment-Scale Snow Water Equivalent Datasets and Models in the Western United States
By: Ritchie et al.

**General Comments**

This manuscript presents a catchment-scale benchmarking of spatially distributed snow water equivalent (SWE) datasets and models across the western United States, using ASO-derived SWE as a reference and the NextGen Hydrofabric as a common spatial support. The authors aggregate gridded SWE products of varying native resolution to hydrofabric catchments using area-weighted remapping and to the full basin scale. They then evaluate performance using a suite of metrics (e.g., rMAE, pbias, KGE, R²) across multiple scenes and physiographic settings. The scope of this analysis is broad, and the dataset-to-dataset comparisons are thoughtfully implemented.

In my view, the strongest contribution of this paper is the inclusion of a large array of widely available SWE products. This has the potential to be one of the most comprehensive catchment-scale evaluations of spatially distributed SWE products to date, and it will be of clear value to both the snow remote sensing and hydrologic modeling communities. The use of a common catchment framework is a defensible and practical way to enable apples-to-apples comparison across products with vastly different native spatial resolutions.

However, the manuscript's repeated framing around protocol standardization is currently unclear and, in places, distracts from the strong scientific results. In my opinion, the paper reads best as a benchmarking study that uses the CHPT framework, rather than as a study that defines a new protocol. I believe this study has the potential to be highly impactful for the snow community once the major concerns below are addressed.

**Major Concerns**

1. **The "protocol standardization" framing is under-defined and appears overstated.**

   The manuscript frequently frames its contribution as the development of a standardized benchmarking protocol, but this framing is under-defined and appears overstated. It is not clear what "protocol" means operationally in this context, including which evaluation elements are fixed versus optional, what default methodological choices are required (e.g., remapping approach, masking, temporal alignment), or how an independent researcher would implement the same protocol in practice. As written, the study appears

to apply an existing CHPT-aligned benchmarking framework to a comprehensive set of SWE products rather than establish a new standard. Reframing the contribution as a near-exhaustive SWE product evaluation that leverages a CHPT-aligned evaluation framework would more accurately reflect the methodological advances presented.

Additionally, prior protocol-focused work under the SnowPEx initiative (Derksen et al., 2015) should be explicitly referenced. SnowPEx was designed to establish methods and protocols for SWE intercomparison and validation, with an emphasis on reproducibility, standardized outputs and metrics, and the use of multiple complementary evaluation activities spanning gridded and watershed-based analyses.

These principles closely align with the stated objectives of this study, particularly its catchment-scale benchmarking using ASO observations. Referencing SnowPEx would strengthen the positioning of the manuscript by situating it within an existing body of protocol-driven SWE evaluation work and by clarifying how the proposed approach builds upon, refines, or diverges from established community frameworks rather than developing them in isolation.

2. **Add analysis with respect to physiographic parameters to help explain better model performance.**

   At present, most performance metrics are reported in an aggregated sense, which can obscure regime-dependent behavior that is known to be important for SWE products. Prior protocol-oriented SnowPEx work emphasizes the need to understand error metrics with respect to various physiographic parameters as they can be closely linked.

   Building on the elevation analysis and reporting model skill with respect to: (1) SWE magnitude, (2) forest fraction, (3) time/seasonality, and (4) topographic complexity would strengthen the interpretability of the results and help ensure that statistics do not mask regime-specific deficiencies that are particularly relevant for mountain water supply and operational applications.

3. **The comparison to CAMELS is conceptually mismatched and should be revised or narrowed.**

   CAMELS was created to enable large-sample hydrology by standardizing climate, streamflow, and catchment attributes across hundreds of basins, making generalization and theory testing possible. This study, in contrast, is primarily a model/product intercomparison built around remapping and evaluation metrics.

   While the motivation for invoking CAMELS is understandable, the comparison currently risks confusing the reader about what is actually novel here. If you want to keep the

reference, I would recommend one sentence on CAMELS as linkage to "the spirit of large-sample hydrology" rather than as a close methodological parallel.

This in no way is a bad thing. You have the scaffolding of a very strong study, but these connections dilute some of the novel scientific findings.

4. **The manuscript should be more explicit about what differentiates this work from prior SWE validation studies.**

   The first goal of the paper appears to be benchmarking, but the emphasis on protocol development creates ambiguity about the core contribution. I recommend the authors state more directly what makes this study different from past modeled/product validation efforts (e.g., SnowPEx, Mudryk et al., Yang et al.). I provide a non-exhaustive list of possible papers in the specific comments.

5. **Uncertainty and potential bias introduced by hydrofabric aggregation should be discussed more explicitly.**

   Aggregating all products to hydrofabric catchments is a reasonable methodological choice given the wide range of native resolutions, but it is not neutral. Catchment aggregation can dampen spatial variability, smooth extremes, and introduce scale-dependent bias, especially when catchments are small relative to coarse products or when terrain and snow heterogeneity within catchments is large.

   A more explicit discussion of how these effects may influence the reported metrics and product rankings would strengthen the interpretation of the results.

6. **SNODAS performance merits further discussion.**

   This is one of the most intriguing and surprising results of this work, that could be discussed/investigated further. I recommend comparing the results presented here to past SNODAS validation work and discussing/hypothesizing why these results show better performance.

In summary, I believe the strong focus on building a community protocol should be reduced. Instead, adding further analysis and discussion would help position this manuscript as one of the most comprehensive spatially distributed validations of SWE products to date, which is a significant and valuable community contribution. I appreciate the motivation to move toward a shared evaluation framework, but I do not think this manuscript is the appropriate venue to formalize it.

If the authors believe that the CHPT standardized protocol is a primary contribution of the manuscript, they must more clearly articulate its novelty with respect to past work and demonstrate how it could be applied in future studies. In addition, they should justify why the specific methodological choices presented here (e.g., hydrofabric aggregation, selected metrics) represent best practice for SWE validation.

As a brief example, hydrofabric aggregation is well suited to this study because of the wide range of dataset resolutions (50 m to >9 km). That said, in other contexts, such as emerging high-resolution SAR-based SWE studies, this type of aggregation could reduce analytical robustness due to spatial averaging. In those cases, alignment with CHPT principles may still be appropriate, but a single community protocol would not necessarily be.

Below is my list of specific comments and technical corrections. Please note that the length/amount of comments directly reflects my desire to see this work become a highly impactful contribution to the snow and hydrology community!

**Specific Comments/Corrections**

1. L27: Provide citation.

2. L28: "Snowmelt volumes can often be predicted with usable accuracy…" is unclear. Referencing discharge? Snowpack remote sensing? What does 'reasonable accuracy' mean?

3. L30: Add reference to the fact snowpack/SWE is highly variable over small spatial scales space (making it hard to measure), mainly driven by physiographic factors. Main points should be: (1) snow is highly variable and hard to measure, (2) this fact complicates streamflow forecasting/water management.

4. L30: Consider using "western United States (WUS)". Additionally, check your formatting is this throughout as it is inconsistent.

5. L30: SWE is defined here, therefore does not to to be again. Check to make sure this is consistent throughout as it is defined and "snow water equivalent" is also used.

6. L31: Officially referred to as, "United States Department of Agriculture's Natural Resources Conservation Service (NRCS) the SNOw TELemetry (SNOTEL) network." Consider a more recent reference for SNOTEL: (Fleming et al., 2023).

7. L33: "spatially distributed" a more common term than "areal"

8. L36: and numbers (1), (2), etc., in front of list items so easily distinguishable to the reader.

9. L38: (1) Remove "dataset" as the line is referring to the Daymet model, (2) Define NLDAS-2.

10. L43: This is the incorrect citation for UCLA-SWE. You should cite both the NSIDC dataset and the data description paper. This also impacts Section 2.2.4 and Table 1, which I will address in more detail below.

    Fang, Y., Liu, Y. & Margulis, S. A. (2022). Western United States UCLA Daily Snow Reanalysis. (WUS_UCLA_SR, Version 1). [Data Set]. Boulder, Colorado USA. NASA National Snow and Ice Data Center Distributed Active Archive Center. https://doi.org/10.5067/PP7T2GBI52I2.

    Fang, Y., Liu, Y., & Margulis, S. A. (2022). A western United States snow reanalysis dataset over the Landsat era from water years 1985 to 2021. *Scientific Data*, *9*(1), 677. https://doi.org/10.1038/s41597-022-01768-7

11. L45: Replace singular Landsat with, "...from Landsat 5–7…" or something similar. Additionally, while Margulis et al. (2019) uses MODIS data, the UCLA-WUS dataset only uses Landsat. This surprised me as well, but I double checked the User Guide (https://nsidc.org/sites/default/files/wus_ucla_sr-v001-userguide.pdf), and only Landsat is referenced for fSCA data. Update accordingly here and throughout the manuscript.

12. L53–54: (1) Move reference ASO after lidar is introduced to make clear they're not the only ones performing that technique,  (2) define passive microwave satellite names if you want to keep them, or just say "passive microwave remote sensing" with a citation, (3) Passive microwave is a direct measurement, update. (4) Consider referencing active radar-based approaches (e.g., InSAR and Ku-/X-band).

13. L54: Just "Airborne lidar…" is sufficient here. Or if you want to keep the lidar definition, which is not needed in most journals anymore (think 'radar'), move to L52 where the term is first used.

14. L55: ASO already defined. Check consistency across the manuscript, as there are other instances of this.

15. L59: Consider a reference to Raleigh & Small (2017) to discuss density modeling uncertainty, as well as discussing this later in the manuscript.

16. L60: Consider framing as, "the most accurate technique for estimating high-resolution basin-scale SWE." Or something similar. Provide citations if you can find one that proves this.

17. L61: "Researchers have duly used ASO datasets to assess remotely sensed snow" Clarify this statement. I think you're trying to get at the fact that ASO has been used as a validation dataset for experimental remote sensing and modeling, but that's opaque in the current framing. Also, Behrangi et al. (2018) assess precipitation products, so I'd move this reference to later in the sentence with those citations.

18. L64: Update, "SWE intercomparisons" to "provide validation for modeled SWE data" or something like that. Given my prior comments about expanding the validation, I would add some text on how this study builds off Yang et al. (2023) to a WUS-wide analysis, as it's the most similar study to what you conducted.

19. L66: Define NWM here as abbreviation is used below.

20. L72: I would add other studies that validate a range of SWE products. A non-exhaustive list of similar SWE validation-type studies to consider, but add whatever you feel is relevant: (Hancock et al., 2013; Kim et al., 2021; Luojus et al., 2021; Mortimer et al., 2022; Mortimer et al., 2024; Ramos Buarque et al., 2025; Zschenderlein et al., 2023).

21. L80–103: Consider removing this text or shortening down to a sentence to streamline the focus of the study.

22. Section 2.1: Add map of ASO basins used in the study. Especially as this was submitted to EGU journal so readers may not be as familiar with WUS geography.

23. L123–132: I would remove this block of text. Can make a note in the appendix if need be, but not pertinent information to the study.

24. Section 2.2: State the three types of products used instead for clarity. Question I'm a bit unclear on: how are reanalysis and model products different in this context?

25. Table 1: Fix Margulis citation and data description. Add m or km value to the Spatial Resolution column for all data entries.

26. L149: Add Raleigh & Small (2017) reference after "density estimates".

27. L149: "<8 cm at 3 m resolution and <2 cm at 50 m" These are the "limited validation" numbers presented by ASO. Could be good to add other references which commonly state higher error values for lidar snow depth, and then add discussion of these uncertainties in the Discussion section.

28. L151: Same comment as #16.

29. L160: Provide citation for RUC2.

30. L161: Add NRCS definition to first SNOTEL reference in L31, see #6.

31. L161–164: Recommend using semicolons to break up the list to mark it very clear each of the agencies or platforms which are providing the data. Reformat so brackets aren't needed for GOES and AVHRR.

32. 2.2.3 Just SWANN is sufficient as the product was not introduced as UA SWE.

33. L170: If COOP is the same as referenced above, move the NWS definition there.

34. L169: "in situ" in EGU journals, check throughout manuscript.

35. L172–174: These lines are unclear to me. It currently reads like it's just a PRISM correction, which I don't think it's the case.

36. L177: Provide more information on what an ANN is here, also defined as ANN is used below.

37. L180: Bring up that SWANN produces two distinct datasets in the beginning of the paragraph.

38. L181: If available, provide the quantitative comparison from Broxton et al. 2024 instead of the qualitative "similar".

39. L182: EGU/TC date formatting: "**Date and time**: 25 July 2007 (dd month yyyy), 15:17:02 (hh:mm:ss). Often it is necessary to specify the time if referring to local time or universal time coordinated. This can be done by adding "LT" or "UTC", respectively. If needed when referring to years, CE (common era) and BCE (before the common era) should be used instead of AD and BC since CE and BCE are more appropriate in interfaith dialogue and science."

40. L184–185: Cite proper paper/dataset here as stated in #10, and update this whole section accordingly. I would recommend including a reference to Margulis et al. (2016), which was the first iteration of this method over the Sierras at 90 m.

41. L191: Fang et al. (2022) is the reference for the correct dataset used in this study, not the HMA version.

42. L192–195: Margulis et al. (2019) refers to the earlier version of the dataset that isn't publicly available. Keep the reference but note it's different from what's used in this work.

43. L198: NWM already introduced.

44. L200: Define what "column land surface model component" is.

45. L203: Provide quantitative values for model performance in the prior studies noted.

46. 2.2.6: Has there been any prior evaluation of this dataset? If so, include values and citations.

47. L208: Define USGS.

48. L212: Define and cite if possible, "Yonsei University planetary boundary layer scheme", and "Rapid Radiative Transfer Model (RRTMG)"

49. L2.2.8: Is this AGU abstract the only available citation for this data? If it's open and available, please create a Zotero or like citation for the data. Also, has there been any prior validation work using this data?

50. L229: Cite the specific Sturm snow classes paper used here – 1995 or 2021.

51. L230: NDSI not NSDI - I mix this up all the time too!!

52. L232: Cite the proper version NLDAS used, likely 2 or 3.

53. L239: Cite Xia et al. (2012) for NLDAS-2.

54. L241: Switch to: Variable Infiltration Capacity (VIC; Liang et al., 1994).

55. L243: Cite NARR.

56. L244: Cite SVAT.

57. 2.2.10: Provide title.

58. L251: Cite Daymet, ie Thorton et al. (2021).

59. L256: Check this citation as Thornton et al. (2000) does not discuss SWE modeling.

60. 2.3: As stated in the Fig. 1 comment (# 63), this section needs more info here on how the coarse resolution data are aggregated. It makes sense for how it works for 50 m data, but curious to how specifically you're dealing with partially covered hydrofabric polygons, especially from the coarse-scale data.

   This is one of the novelties of this work that needs to be expanded on, specifically not just the package names, but the information on processing techniques they're performing under the hood.

61. Fig. 1: I'm a bit confused how the coarser datasets are averaged to this small scale grid. For example, how does this process work for ERA5-Land? I think adding additional rows with both a moderate resolution dataset (500–1000 m) and a coarse one (9 km?) would better illuminate the processes to the reader.

62. Section 2.2.4: If one of the goals is still to create "community protocol", please provide evidence of what these are the optimal metrics to evaluate the data with. Recent work from Clark et al. (2021) and Williams (2025) discuss the overuse of KGE for hydrologic model evaluation. While a different application than this analysis, I give this example to exemplify the importance of justifying validation metric selection.

63. L283: Explain what KGE is and what utility it provides, as it's not as common as the other metrics.

64. L283: "Metrics were calculated across all catchments in a scene, yielding one value per metric per scene." Does this mean per hydrofabric catchment? I'd recommend stating that you will refer to each hydrofabric polygon/catchment as simply 'catchment' earlier in the manuscript somewhere.

65. L296–298: Consider removing CAMELS reference here per reasons stated in my early comments.

66. L301–306: Considering removing this text in accordance with my early comments.

67. Table 2: Move this to Appendix or remove. It's unclear to me what this is offering to the research, as it's just common/basic principles of performing a robust scientific study.

68. L315–318: Might need to update this text to represent the further analysis added to the study.

69. L329–336: Provide quantitative values for your statements like you did in the previous paragraph, here and throughout the results section. Makes the comparison much more intuitive for the reader instead of just using qualitative descriptions like "more", "similar", and "less".

70. L341: Move this KGE description to Section 2.4 as noted in #63, and add more detail to what this metric is telling us about model performance.

71. Figure 4: State that medians are shown by the vertical dotted lines with their value denoted in the legend.

72. L353–356: Good text, but move to the discussion section.

73. L356–365: See #69.

74. Figure 5: This analysis is not described in the methods section. Add that and describe what the specified elevation bands were chosen. A note for later as well, some discussion of the importance of elevation as it relates to total SWE and its importance to streamflow is warranted.

75. Section 4.3: Add average metrics across the regions for comparison. Also rMAE is used earlier, but MAE is reported here. Is there a reason for that?

76. Figure 6: Reformat graph so it's not flipped and rearrange the plots so the CA vs. CO graphs for each dataset are next to each other for better comparison. I'd also standardize the y-axis to 3000mm so they can be compared apples-to-apples.

77. Section 4.4: See #63.

78. L394: Switch ⅛ degree to approximate value in meters.

79. Figure 7: This figure is hard to read as many of the points overlap. I think a heatmap table where cell values are scaled by color shade (i.e., dark red means bad worse error metric), would be much easier for the reader to understand.

80. L403: "...potential application of this standardized approach for benchmarking a range of products…" I'm still unclear of what this standardized approach is, consider removing.

81. Section 5.1: Broadly, this section could benefit from discussion of how this work compares to past studies using these datasets.

82. L438: "We find that operational products are not notably worse than non-operational products, given that non-operational products such as CONUS404, UCLA-SWE, and ERA5-Land exhibited mixed or inferior performance to SNODAS."

   This statement is broad and would benefit from clearer support in the results. Are you referencing the 'catchment' analysis? Basin-SWE average? What does "not noticeably worse" mean? (which could be changed to 'performed similarly'). What was this mixed or inferior performance? How does that compare to CO vs. CA? If fully justified, this is a super interesting result that should be expanded upon.

83. L438: "While performance varies across operational and non-operational products, strong results from UCLA-SWE and SNODAS show that operational products can approach retrospective reanalyses in quality."

   If you are going to assert this, I would present an analysis which shows average performance between operational vs. non-operational products. Right now you're just stating this is true, but not presenting results that clearly show this.

84. L446: Consider adding Subsection for the regional analysis discussion.

85. L464–467: Use these lines as the starting point of a new subsection about study uncertainties. Some but not all questions to consider:

   (1) what are the uncertainties associated with ASO SWE and how would those impact the results of the study?

   (2) How does the use of averaging to the hydrofabric catchment impact results?

   For example, a given 'catchment' could be massively overestimating SWE in one area, underestimating in another, but the average comes out to almost exactly

what ASO's average is. Your method would show the model performing well, yet in reality it's not properly representing the snowpack. Consideration of this assumption should be addressed fully.

86. Section 5.2: I disagree that a "community experimental protocol" was established in this study, as stated in my prior comments. Consider removing or shortening.

87. Section 5.2.1: A few thoughts here: I'd recommend removing this section in favor of adding to the Results and Discussion section with the analysis described in the General Comments. That said, if you can justify why it should be kept, it needs to be moved into the Results section and described in the Methods, not just randomly presented in the Discussion section of the paper.

I'm a bit confused as to why this is first introduced here and what it adds. To me, it seems you're just performing a smaller analysis on data you've already analyzed? Am I correct to say these datasets were technically included in the main analysis of the study?

88. L520: "...further outlined and demonstrated an example of a community evaluation experimental protocol drawing from the datasets used in this study." Check this line if you decide to remove Section 5.2.1.

89. L529–534: Move to discussion! I was looking for some text like this there.

90. L522–529: This is a fantastic summary of the analysis!

91. L535–551: Consider removing this section if you deemphasize focus on creating "community protocol".

**References**

Clark, M. P., Vogel, R. M., Lamontagne, J. R., Mizukami, N., Knoben, W. J. M., Tang, G., et al. (2021). The Abuse of Popular Performance Metrics in Hydrologic Modeling. *Water Resources Research*, *57*(9), e2020WR029001. https://doi.org/10.1029/2020WR029001

Derksen, C., Brown, R., Mudryk, L., Luojus, K., & Vuyovich, C. (2015). Methods and Protocols for Intercomparing and Validating Snow Extent and Snow Water Equivalent products.

Fleming, S. W., Zukiewicz, L., Strobel, M. L., Hofman, H., & Goodbody, A. G. (2023). SNOTEL, the Soil Climate Analysis Network, and water supply forecasting at the Natural Resources Conservation Service: Past, present, and future. *JAWRA Journal of the American Water Resources Association*, 1752–1688.13104. https://doi.org/10.1111/1752-1688.13104

Hancock, S., Baxter, R., Evans, J., & Huntley, B. (2013). Evaluating global snow water equivalent products for testing land surface models. *Remote Sensing of Environment*, *128*, 107–117. https://doi.org/10.1016/j.rse.2012.10.004

Kim, R. S., Kumar, S., Vuyovich, C., Houser, P., Lundquist, J., Mudryk, L., et al. (2021). Snow Ensemble Uncertainty Project (SEUP): quantification of snow water equivalent uncertainty across North America via ensemble land surface modeling. *The Cryosphere*, *15*(2), 771–791. https://doi.org/10.5194/tc-15-771-2021

Luojus, K., Pulliainen, J., Takala, M., Lemmetyinen, J., Mortimer, C., Derksen, C., et al. (2021). GlobSnow v3.0 Northern Hemisphere snow water equivalent dataset. *Scientific Data*, *8*(1), 163. https://doi.org/10.1038/s41597-021-00939-2

Margulis, S. A., Cortés, G., Girotto, M., & Durand, M. (2016). A Landsat-Era Sierra Nevada Snow Reanalysis (1985–2015). *Journal of Hydrometeorology*, *17*(4), 1203–1221. https://doi.org/10.1175/JHM-D-15-0177.1

Mortimer, C., Mudryk, L., Derksen, C., Brady, M., Luojus, K., Venäläinen, P., et al. (2022). Benchmarking algorithm changes to the Snow CCI+ snow water equivalent product. *Remote Sensing of Environment*, *274*, 112988. https://doi.org/10.1016/j.rse.2022.112988

Mortimer, Colleen, Mudryk, L., Cho, E., Derksen, C., Brady, M., & Vuyovich, C. (2024). Use of multiple reference data sources to cross-validate gridded snow water equivalent products over North America. *The Cryosphere*, *18*(12), 5619–5639. https://doi.org/10.5194/tc-18-5619-2024

Raleigh, M. S., & Small, E. E. (2017). Snowpack density modeling is the primary source of uncertainty when mapping basin-wide SWE with lidar: Uncertainties in SWE Mapping With Lidar. *Geophysical Research Letters*, *44*(8), 3700–3709. https://doi.org/10.1002/2016GL071999

Ramos Buarque, S., Decharme, B., Barbu, A. L., & Franchisteguy, L. (2025). Insights into the North Hemisphere daily snowpack at high resolution from the new Crocus–ERA5 product. *Earth System Science Data*, *17*(12), 7227–7249. https://doi.org/10.5194/essd-17-7227-2025

Williams, G. P. (2025). Friends don't let friends use Nash-Sutcliffe Efficiency (NSE) or KGE for hydrologic model accuracy evaluation: A rant with data and suggestions for better practice. *Environmental Modelling & Software*, *194*, 106665. https://doi.org/10.1016/j.envsoft.2025.106665

Xia, Y., Mitchell, K., Ek, M., Sheffield, J., Cosgrove, B., Wood, E., et al. (2012). Continental-scale water and energy flux analysis and validation for the North American Land Data Assimilation System project phase 2 (NLDAS-2): 1. Intercomparison and application of model products: WATER AND ENERGY FLUX ANALYSIS. *Journal of Geophysical Research: Atmospheres*, *117*(D3), n/a-n/a. https://doi.org/10.1029/2011JD016048

Zschenderlein, L., Luojus, K., Takala, M., Venäläinen, P., & Pulliainen, J. (2023). Evaluation of passive microwave dry snow detection algorithms and application to SWE retrieval during seasonal snow accumulation. *Remote Sensing of Environment*, *288*, 113476. https://doi.org/10.1016/j.rse.2023.113476